# DOES CONTINUAL LEARNING EQUALLY FORGET ALL PARAMETERS?

## ABSTRACT

Distribution shift (e.g., task or domain shift) in continual learning (CL) usually results in catastrophic forgetting of neural networks. Although it can be alleviated by repeatedly replaying buffered data, the every-step replay is time-consuming and the memory to store historical data is usually too small for retraining all parameters. In this paper, we study which modules in neural networks are more prone to forgetting by investigating their training dynamics during CL. Our proposed metrics show that only a few modules are more task-specific and sensitively alters between tasks, while others can be shared across tasks as common knowledge. Hence, we attribute forgetting mainly to the former and find that finetuning them only on a small buffer at the end of any CL method can bring non-trivial improvement. Due to the small number of finetuned parameters, such "Forgetting Prioritized Finetuning (FPF)" is efficient on both the computation and buffer size required. We further propose a more efficient and simpler method that entirely removes the every-step replay and replaces them by only $k$-times of FPF periodically triggered during CL. Surprisingly, this "$k$-FPF" performs comparably to FPF and outperforms the SOTA CL methods but significantly reduces their computational overhead and cost. In experiments on several benchmarks of class- and domain-incremental CL, FPF consistently improves existing CL methods by a large margin and $k$-FPF further excels on the efficiency without degrading the accuracy. We also empirically studied the impact of buffer size, epochs per task, and finetuning modules to the cost and accuracy of our methods.

## 1 INTRODUCTION

Empowered by advancing deep learning techniques and neural networks, machine learning has achieved unprecedented promising performance on challenging tasks in different fields, mostly under the i.i.d. (independent and identically distributed) offline setting. However, its reliability and performance degenerates drastically in continual learning (CL) where the data distribution or task in training is changing over time, as the model quickly adapts to a new task and overwrites the previously learned weights. This leads to severe bias towards more recent tasks and "catastrophic forgetting" of previously learned knowledge, which is detrimental to a variety of practical applications.

A widely studied strategy to mitigate forgetting is experience replay (ER) (Ratcliff, 1990; Robins, 1995) and its variants (Riemer et al., 2018; Buzzega et al., 2020; Boschini et al., 2022), which store a few data from previous tasks in a limited memory and train the model using both the current and buffered data. However, they only bring marginal improvements when the memory is too small to store sufficient data for recovering previously learned knowledge, which is common due to the complicated distributions of previous tasks. In contrast, multi-task learning (Caruana, 1997) usually adopts a model architecture composed of a task-agnostic backbone network and multiple task-specific adaptors on top of it. While the backbone needs to be pre-trained on large-scale data, the adaptors are usually light-weight and can be achieved using a few data. In CL, however, we cannot explicitly pre-define and separate the task-agnostic parts and task-specific parts. Although previous methods (Schwarz et al., 2018; Zenke et al., 2017) have studied to restrict the change of parameters critical to previous tasks, such extra constraint might degrade the training performance and discourage task-agnostic modules capturing shared knowledge.

In this paper, we study a fundamental but open problem in CL, i.e., are most parameters task-specific and sensitively changing with the distribution shift? Or is the catastrophic forgetting mainly caused by the change on a few task-specific parameters? It naturally relates to the plasticity-stability trade-off in biological neural systems (Mermillod et al., 2013): more task-specific parameters improves the plasticity but may cause severe forgetting, while the stability can be improved by increasing parameters shared across tasks. In addition, how many task-specific parameters suffice to achieve promising performance on new task(s)? Is every-step replay necessary?

To answer these questions, we investigate the training dynamics of model parameters during the course of CL by measuring their changes over time. For different CL methods training with various choices of buffer size and number of epochs per task on different neural networks, we consistently observe that **only a few parameters change more drastically than others between tasks**. The results indicate that most parameters can be shared across tasks and **we only need to finetune a few task-specific parameters to retain the previous tasks' performance**. Since these parameters only contain a few layers of various network architectures, they can be efficiently and accurately finetuned using a small buffer.

The empirical studies immediately motivate a simple yet effective method, "forgetting prioritized finetuning (FPF)", which finetunes the task-specific parameters using buffered data at the end of CL methods. Surprisingly, on multiple datasets, FPF consistently improves several widely-studied CL methods and substantially outperforms a variety of baselines. Moreover, we extend FPF to a more efficient **replay-free CL method** "$k$-FPF" that entirely eliminates the cost of every-step replay by replacing such frequent replay with occasional FPF. $k$-FPF applies FPF only $k$ times during CL. We show that a relatively small $k$ suffices to enable $k$-FPF achieving comparable performance with that of FPF+SOTA CL methods and meanwhile significantly reduces the computational cost. In addition, we explore different groups of parameters to finetune in FPF and $k$-FPF by ranking their sensitivity to task shift evaluated in the empirical studies. For FPF, we compare them under different choices for the buffer size, the number of epochs per task, the CL method, and the network architecture. FPF can significantly improve existing CL methods by only finetuning $\leq 0.127\%$ parameters. For $k$-FPF, we explore different groups of parameters, $k$, and the finetuning steps per FPF. $k$-FPF can achieve a promising trade-off between efficiency and performance. Our experiments are conducted on a broad range of benchmarks for class- and domain-incremental CL in practice, e.g., medical image classification and realistic domain shift between image styles.

## 2 RELATED WORK

**Continual Learning and Catastrophic Forgetting** A line of methods stores samples of past tasks to combat the forgetting of previous knowledge. ER (Riemer et al., 2018) applies reservoir sampling (Vitter, 1985) to maintain a memory buffer of uniform samples over all tasks. Each mini-batch of ER is randomly sampled from current task and the buffered data. MIR (Aljundi et al., 2019) proposes a new strategy to select memory samples suffering the largest loss increase induced by the incoming mini-batch so those at the forgetting boundary are selected. DER and DER++ (Buzzega et al., 2020) apply knowledge distillation to mitigate forgetting by storing the output logits for buffered data during CL. iCaRL (Rebuffi et al., 2017) selects samples closest to the representation mean of each class and trains a nearest-mean-of-exemplars classifier to preserve the class information of samples. A-GEM (Chaudhry et al., 2018) constrains new task's updates to not interfere with previous tasks. Our methods are complementary techniques to these memory-based methods. It can further improve their performance by finetuning a small portion of task-specific parameters on buffered data once (FPF) or occasionally ($k$-FPF).

Another line of work imposes a regularization on model parameters or isolates task-specific parameters to retain the previous knowledge. oEWC (Schwarz et al., 2018) constrains the update of model parameters important to past tasks by a quadratic penalty. To select task-specific parameters, SI (Zenke et al., 2017) calculates the effect of the parameter change on the loss while MAS (Aljundi et al., 2018) calculates the effect of parameter change on the model outputs when each new task comes. PackNet (Mallya & Lazebnik, 2018) and HAT (Serra et al., 2018) iteratively assign a subset of parameters to consecutive tasks via binary masks. All these works try to identify critical parameters for different tasks during CL and restrict the update of these parameters. But they can also prevent task-agnostic parameters from learning shared knowledge across tasks. From the

training dynamics of CL, we identify the parameters sensitive to distribution shift. FPF and $k$-FPF finetune these parameters to mitigate bias without restricting the update of task-agnostic parameters.

**Different modules in neural networks** (Ramasesh et al., 2020) shows that freezing earlier layers after training the first task have little impact on the performance of the second task. This is because their unfrozen part covers the last FC layer and many BN parameters, which are the most sensitive/critical according to our empirical study. Moreover, they did not take into account that the earlier layers have much less parameters and capacity than the top layers. (Pham et al., 2022) only studies the effect of different normalization layers on CL while our method investigates the sensitivity of all parameters in different network architectures. Their continual-norm still suffers from the forgetting to task shift. Our methods directly finetune the task-specific layers on the buffered data to eliminate the bias caused by the task drift. (Zhang et al., 2019) argues different layers play different roles in the representation function. They find that in different architectures, the parameters in the top layers(close to input) are more critical and perturbing them leads to poor performance. Our empirical study is consistent with their findings in that the earlier convolutional layer is sensitive to task drift and the induced biases on them lead to catastrophic forgetting.

## 3   PROBLEM SETUP

**Notations** We consider the CL setting, where the model is trained on a sequence of tasks indexed by $t \in \{1, 2, \ldots, T\}$. During each task $t$, the training samples $(x, y)$ (with label $y$) are drawn from an i.i.d. distribution $D_t$. Given a neural network $f_\Theta(\cdot)$ of $L$ layers with parameter $\Theta = \{\theta_\ell\}_{\ell=1:L}$, $\theta_\ell = \{\theta_{\ell,i}\}_{i=1:n_\ell}$ denote all parameters in layer-$\ell$ where $\theta_{\ell,i}$ denotes parameter-$i$. On each task, $f_\Theta(\cdot)$ is trained for $N$ epochs. We denote all parameters and the layer-$\ell$'s parameters at the end of the $n$-th epoch of task $t$ by $\Theta_n^t$ and $\theta_{\ell,n}^t$, $n \in \{1, \ldots, N\}$, respectively.

**Settings** In this paper, we mainly focus on class-incremental learning (class-IL) and domain-incremental learning (domain-IL). In class-IL, $D_t$ are drawn from a subset of classes $C_t$ and $\{C_t\}_{t=1}^T$ for different tasks are assumed to be disjoint. class-IL is a more challenging setting of CL(Van de Ven & Tolias, 2019) than task-incremental learning (task-IL) (Lopez-Paz & Ranzato, 2017). Unlike task-IL, class-IL cannot access to the task label during inference and has to distinguish among all classes from all tasks. In domain-IL, tasks to be learnt remain the same but the domain varies, i.e. the input data distribution $D_t$ changes. The model is expected to adapt to the new domain without forgetting the old ones. The goal of the class-IL and domain-IL is: $\min_\Theta L(\Theta) \triangleq \sum_{t=1}^T \mathbb{E}_{(x,y) \sim D_t}[l(y, f_\Theta(x))]$, where $l$ is the objective function.

**Class-IL datasets** We conduct class-IL experiments on Seq-MNIST, Seq-OrganAMNIST, Seq-PathMNIST, Seq-CIFAR-10, and Seq-TinyImageNet. Seq-OrganAMNIST and Seq-PathMnist are generated by splitting OrganAMNIST or PathMNIST from MedMNIST(Yang et al., 2021), a medical image classification benchmark. CL on medical images is important in practice but also challenging since medical images always come as a stream with new patients and new deceases. Moreover, medical images of different classes might only have subtle differences that are hard to distinguish. Both Seq-OrganAMNIST and Seq-PathMnist consist of 4 disjoint classification tasks. The number of classes per task in Seq-OrganAMNIST and Seq-PathMnist are [3, 3, 3, 2] and [3, 2, 2, 2] respectively. Seq-MNIST (Seq-CIFAR-10) are generated by splitting the 10 classes in MNISTLeCun et al. (1998) (CIFAR-10Krizhevsky et al. (2009)) into five binary classification tasks. Seq-TinyImageNet partitions the 200 classes of TinyImageNet(Le & Yang, 2015) into 10 disjoint classification tasks with 20 classes per task.

**Domain-IL datasets** We conduct domain-IL experiments on PACS dataset (Li et al., 2017), which is widely used for domain generalization. It can present more realistic domain-shift challenge than the toy-setting of PermuteMNIST (Kirkpatrick et al., 2017). Images in PACS come from seven classes and belong to four domains: Paintings, Photos, Cartoons, and Sketches. In Seq-PACS for CL, each task only focuses on one domain and the sequence of tasks is Sketches → Cartoons → Paintings → Photos (increasing the level of realism over time) (Volpi et al., 2021).

**Models** We follow the standard network architectures adopted in most previous CL works. For Seq-MNIST, following Lopez-Paz & Ranzato (2017); Riemer et al. (2018), we employ an MLP, i.e., a fully-connected (FC) network with two hidden layers, each composed of 100 ReLU units. Following (Rebuffi et al., 2017; Li et al., 2020; Derakhshani et al., 2022), we train ResNet-18 (He

et al., 2016) on other five datasets. In addition, we also extend our empirical study to another architecture, i.e., VGG-11 (Simonyan & Zisserman, 2014) on Seq-CIFAR-10.

## 4    FORGETTING OF DIFFERENT PARAMETERS: AN EMPIRICAL STUDY

A fundamental and long-lasting question in CL is how the distribution shift impacts different model parameters and why it leads to harmful forgetting. Its answer could unveil the plasticity-stability trade-off in CL, where some parameters are plastic and task-specific and thus have to be finetuned before deploying the model, while the stable ones can be shared with and generalized to new tasks. In order to answer this question, we conduct an comprehensive empirical study that compares the training dynamics of different parameters in three widely studied neural networks. We propose a novel metric measuring the sensitivity of parameters to distribution shifts. On all the studied networks, it helps us distinguish between plastic and stale parameters and allocate the task-specific ones.

### 4.1    MEASURING FORGETTING VIA TRAINING DYNAMICS

To measure and compare the forgetting effects of different parameters, we adopt an intuitive metric to compute the change of parameters and investigate their dynamics over CL. In CL, the unstable changes of parameters are mainly caused by the task shift, while the learning within each task usually leads to smooth changes. Hence, the proposed metric focuses on the difference between two consecutive tasks, e.g., the change of parameters between epoch-$n$ of the two tasks, i.e., $(1/|\theta_\ell|)\|\theta_{\ell,n}^{t+1} - \theta_{\ell,n}^t\|_1$. Its results on different neural networks are displayed in Fig. 1.

### 4.2    FORGETTING OF DIFFERENT PARAMETERS DURING CL

We first investigate and compare the training dynamics of different parameters in three types of neural networks. To gain insights applicable to all CL methods, we exclude any specific CL techniques but simply apply SGD to train a model on a sequence of tasks, without any countermeasure to forgetting. Then, we extend the experiment to different CL methods, hyper-parameters (e.g., buffer size), and datasets to verify whether the observations still hold.

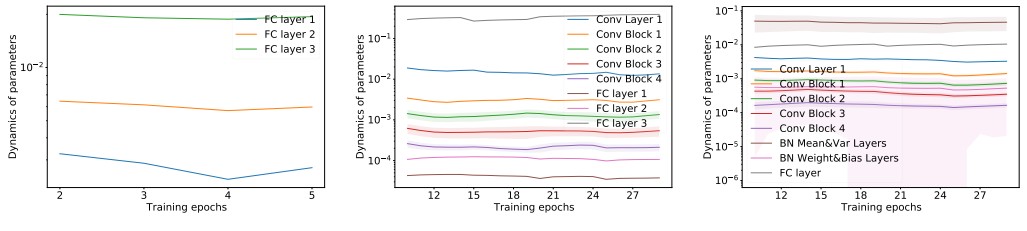

| (a) Dynamics of MLP | (b) Dynamics of VGG-11 | (c) Dynamics of ResNet-18 |

Figure 1: The training dynamics of different groups of parameters when applying SGD in CL to train three types of deep neural networks. Note the the y-axis is of logarithmic scale.

**Dynamics of MLP** We train a three-layer MLP (including the classifier) for Seq-MNIST. Fig. 1(a) reports how the metric introduced in Section 4.1 for each layer changes in CL. It shows that the top FC layer (closest to the output) is the most sensitive one with the greatest changes among all the three layers. This is because tasks in class-IL differ on their predicted classes, which are the outputs of the FC layer. Since task shift mainly changes the top FC layer, finetuning it using all-tasks' data help reduce the forgetting.

**Dynamics of VGG** Fig. 1 (b) shows the training dynamics of parameters in VGG-11 when trained on Seq-CIFAR10. We partition all parameters into several groups, i.e., the bottom convolutional layer (closest to the input), convolutional layers in different blocks, and three top FC layers. The observations on FC layers are consistent with those on MLP: the last FC layer in VGG is much more sensitive to the task shift than other two FC layers. In contrast, the sensitivity of all convolutional layers increases as the layer becomes closer to the input because they are producing the representations for the input images, whose distribution shift directly impacts the bottom convolutional layer. However, they are still more stable (or less plastic) than the last FC layer. Since task shift mainly changes the bottom convolutional layers (among all convolutional layers), finetuning them can be important to alleviate the forgetting.

**Dynamics of ResNet** Fig. 1 (c) reports the training dynamics of parameters in ResNet-18 when trained on Seq-CIFAR10. In addition to the groups of VGG-11, ResNet-18 applies batch-normalization (BN) layers, which have two groups of parameters, i.e., (1) their weights and bias, and (2) their mean and variance. Unlike MLP or VGG-11, in ResNet-18, BN layers' mean and variance become the most changed parameters. This observation makes intuitive sense because the mean and variance of BN layers capture the first and second order moments of the distribution for the latent representations. Except BN mean and variance, the last FC layer and the bottom convolutional layers are still the top-2 sensitive groups among the rest parameters, which are consistent with the observations on MLP and VGG-11. The variance of BN weight and bias is relatively large compared to the rest layers, please refer to Appendix for dynamics of BN weight and bias in different groups.

In the above section, we observe that for three types of deep neural networks, parameters are not equally sensitive to the distribution shift in CL. Moreover, only a small portion of them are much more sensitive and task-specific than others. This implies that only finetuning these task-specific (or plastic) parameters may suffice to retain the previous tasks. That being said, the empirical study is limited to SGD without applying any other CL techniques and only focuses on class-IL. In the following, we extend the studies to different CL methods, buffer sizes, non-standard datasets, and domain-IL, while fixing the model to be ResNet-18.

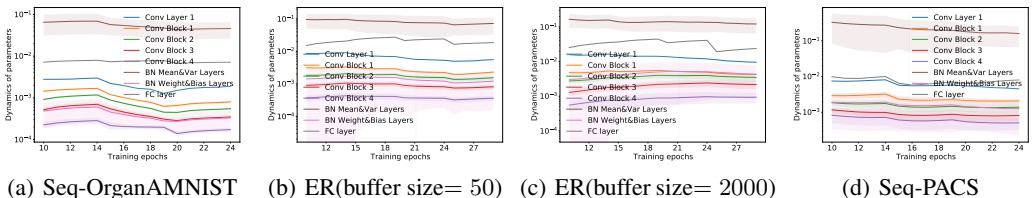

(a) Seq-OrganAMNIST  (b) ER(buffer size= 50)  (c) ER(buffer size= 2000)  (d) Seq-PACS

Figure 2: The training dynamics of different groups of parameters in ResNet-18: (a) on a non-standard dataset; (b,c) using a different CL method with different buffer sizes; (d) in domain-IL setting. Note the the y-axis is of logarithmic scale.

**Dynamics on different scenarios** Fig. 2 (a) extends the empirical study to a medical image dataset Seq-OrganAMNIST. Comparing to Seq-CIFAR-10, it differs on the number of tasks, dataset size, image size, and data type. Despite these differences, the sensitive groups of parameters stays the same. We further replace SGD with ER using two replay buffer sizes, whose results are reported in Fig. 2(b)-(c). The ranking order of parameter groups in terms of sensitivity stays consistent under the change of the replay strategy and buffer size.

**Dynamics on domain-IL** In domain-IL, as shown in Fig. 2 (d), the training dynamics of different parameters is in line with our observations in class-IL: only a small portion of parameters are task-specific. However, one difference is worth noting. Since the output classes stay the same across tasks and only the input domain changes, the most sensitive parameters in class-IL, i.e., the last FC layer, becomes equally or less sensitive than the bottom convolutional layer. Hence, the plasticity and stability of parameters are impacted by how close they are to the changed data distributions.

## 5 FORGETTING PRIORITIZED FINETUNING (FPF) METHODS

The above empirical study of the training dynamics on parameters immediately motivates a simple but novel method for CL, i.e., "forgetting prioritized finetuning (FPF)", which can be applied to any existing CL method. In the more efficient $k$-FPF, we further remove the every-step replay and any other CL techniques but simply applies $k$-times of FPF in SGD training. In Fig. 3, we provide an illustration that compares SGD, replay-based methods and our methods.

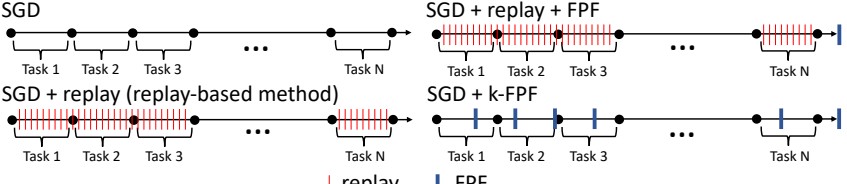

Figure 3: Comparison of SGD, replay-based method, FPF and $k$-FPF. SGD trains each task sequentially without replay. Replay-based methods train model on buffered data and current data simultaneously. FPF finetunes the most sensitive (plastic) parameters for a few iterations using the buffered data at the end of arbitrary CL methods. $k$-FPF periodically (regardless of task boundaries) applies FPF for $k$ times over the course of training.

**FPF to improve CL performance.** FPF applies light-weight finetuning to the most task-specific parameters using the buffered data after the training of arbitrary CL methods. Hence, it is complementary to any existing CL methods as a correction step to remove their biases in the task-specific parameters by finetuning them on the unbiased buffered data. Thereby, it can improve the performance of any existing CL methods without causing notably extra computation.

$k$**-FPF to improve CL efficiency and performance.** FPF is a simple technique that brings non-trivial improvement but it is applied after the training of an existing CL method. Unfortunately, many SOTA CL methods require time-consuming replay in every step, which at least doubles the total computation. Since only a few parameters are sensitive during the task shift, can we develop a replay-free and lazy CL that replaces every-step-replay with occasional FPF? We propose $k$-FPF that applies FPF $k$ times during CL as shown in Fig. 3. Without the costly experience replay, $k$-FPF can still achieve comparable performance as FPF+SOTA CL methods but only requires nearly half of their computation. We can apply $k$-FPF with any replay-free method, e.g., SGD, which is usually used as a lower-bound for CL methods. We still maintain a small buffer by reservoir sampling but it is only for FPF so SGD never accesses it. We lazily apply FPF on the buffer after every $\tau$ SGD steps (in total $k$ times over $k\tau$ SGD steps) without knowing the task boundaries.

$k$**-FPF-CE+SGD** We propose two variants of $k$-FPF, i.e., $k$-FPF-CE+SGD and $k$-FPF-KD+SGD. $k$-FPF-CE+SGD uses the cross-entropy loss to update the sensitive parameters during FPF. In this paper, $k$-FPF-CE refers to $k$-FPF-CE+SGD if not specified. The objective of FPF in $k$-FPF-CE is: $\min_{\Theta^\star} L(\Theta^\star) \triangleq \mathbb{E}_{(x,y) \sim B}[l_{CE}(y, f_\Theta(x))]$ where $\Theta^\star$ denotes selected groups of task-specific parameters, $B$ refers to the buffered data and $l_{CE}$ is the cross-entropy loss.

$k$**-FPF-KD+SGD to further improve performance** Inspired by DER (Buzzega et al., 2020), we further propose $k$-FPF-KD that introduces knowledge distillation (KD) (Hinton et al., 2015) to the objective in $k$-FPF-CE. In this paper, $k$-FPF-KD refers to $k$-FPF-KD+SGD if not specified. Same as DER, the pre-softmax responses (i.e. logits) for buffered data at training time are stored in buffer as well. During FPF, the current model is trained to match the buffered logits to retain the knowledge of previous models. The objective of FPF in $k$-FPF-KD is: $\min_{\Theta^\star} L(\Theta^\star) \triangleq \mathbb{E}_{(x,y) \sim B}[l_{CE}(y, f_\Theta(x))] + \lambda \mathbb{E}_{(x,z) \sim B}[l_{MSE}(z, h_\Theta(x))]$ where $z$ is the logits of buffered sample $x$, $l_{MSE}$ refers to the mean-squared loss, $h_\Theta(x)$ computes the pre-softmax logits and $\lambda$ is a hyper-parameter balancing the two terms. Compared to the computation of $k$-FPF-CE, the additional computation by $k$-FPF-KD is negligible.

**Selection of sensitive parameters for FPF and $k$-FPF**

A key challenge in both FPF and $k$-FPF is to select the task-specific parameters for finetuning. Please refer to A.1 for the detailed metric of selecting sensitive parameters in different neural networks. In the experiments later, under different scenarios and on various benchmarks, we evaluate the performance of FPF and $k$-FPF when selecting different subsets of task-specific parameters according to the training dynamics studies in empirical study. In a nutshell, finetuning more sensitive parameters achieve more improvement, which is in line with our findings in empirical studies. For FPF, finetuning one or two the most sensitive groups of layers, like the last FC layer, is enough to achieve the best performance among all evaluated combinations. In all scenarios, FPF consistently improves CL's performance by a large margin. For $k$-FPF, finetuning slightly more parameters, i.e., the earlier convolutional layers, achieves the best performance, which is comparable with that of FPF+CL. This is a price of removing replay, which halves the computational cost.

# 6 EXPERIMENTS

In this section, to compare FPF and $k$-FPF with SOTA CL methods, we conduct our experiments mainly on ResNet-18. Please refer to the Appendix for the results on other neural networks. We apply FPF and $k$-FPF to multiple benchmark datasets and compare them with SOTA CL baselines in terms of test accuracy and efficiency. Besides, we also compare the performance of finetuning different parameters in FPF and $k$-FPF and show that finetuning a small portion of task-specific parameters suffices to improve CL. FPF improves SOTA CL methods by a large margin under all these scenarios while $k$-FPF achieves comparable performance with FPF but is more efficient.

Table 1: Test accuracy (%) of CL baselines, FPF and $k$-FPF. "-" indicates that the algorithm is not applicable to the setting. For FPF and $k$-FPF, we report the best performance among all combinations of parameters in Fig. 5. $k$-FPF-KD applies an additional knowledge distillation loss to the finetuning objective of $k$-FPF-CE. **Bold** and Bold gray mark the best and second best accuracy.

| BUFFER | METHODS | CLASS-IL | | | | DOMAIN-IL |
| | | SEQ-ORGANAMNIST | SEQ-PATHMNIST | SEQ-CIFAR-10 | SEQ-TINY-IMAGENET | SEQ-PACS |
|---|---|---|---|---|---|---|
| | JOINT | 91.92±0.46 | 82.47±2.99 | 81.05±1.67 | 41.57±0.55 | 70.85±8.90 |
| | SGD | 24.19±0.15 | 23.65±0.07 | 19.34±0.06 | 7.10±0.14 | 31.43±6.39 |
| | oEWC (SCHWARZ ET AL., 2018) | 22.71±0.67 | 22.36±1.18 | 18.48±0.71 | 6.58±0.12 | 35.96±4.59 |
| | GDUMB (PRABHU ET AL., 2020) | 61.78±2.21 | 46.31±5.64 | 30.36±2.65 | 2.43±0.31 | 34.16±3.45 |
| | $k$-**FPF-CE** | 75.21±2.03 | 72.88±3.22 | 57.97±1.53 | 13.76±0.72 | 60.70±2.81 |
| | $k$-**FPF-KD** | 80.32±1.16 | **74.68±4.72** | 58.50±1.03 | 14.74±0.94 | 63.15±1.19 |
| | ER (RIEMER ET AL., 2018) | 71.69±1.71 | 51.66±5.86 | 45.71±1.44 | 8.15±0.25 | 51.53±5.10 |
| | **FPF+ER** | 77.66±1.93 | 67.34±2.68 | 57.68±0.76 | 13.13±0.63 | 65.16±1.97 |
| 200 | AGEM (CHAUDHRY ET AL., 2018) | 24.16±0.17 | 27.93±4.24 | 19.29±0.04 | 7.22±0.15 | 40.54±3.43 |
| | **FPF+AGEM** | 73.76±2.45 | 67.04±4.51 | 55.40±1.97 | 13.24±0.54 | 57.33±0.76 |
| | iCARL (REBUFFI ET AL., 2017) | 79.61±0.56 | 54.35±0.94 | 59.60±1.06 | 12.13±0.20 | - |
| | **FPF+iCARL** | 80.24±0.70 | 71.83±1.51 | **63.95±0.84** | **17.45±0.38** | - |
| | FDR (BENJAMIN ET AL., 2018) | 68.29±3.27 | 44.27±3.20 | 41.77±4.24 | 8.81±0.19 | 45.91±3.54 |
| | **FPF+FDR** | 76.92±1.38 | 70.08±4.06 | 52.49±2.97 | 12.25±0.77 | 58.38±1.70 |
| | DER (BUZZEGA ET AL., 2020) | 73.28±1.33 | 54.45±5.92 | 47.04±3.03 | 9.89±0.58 | 46.93±4.94 |
| | **FPF+DER** | 79.63±1.21 | 67.29±3.75 | 57.25±2.19 | 12.62±1.08 | 61.49±1.37 |
| | DER++ (BUZZEGA ET AL., 2020) | 78.22±2.05 | 62.00±3.79 | 59.13±0.81 | 12.12±0.69 | 55.75±2.02 |
| | **FPF+DER++** | **80.99±0.91** | 68.78±2.89 | 61.98±1.04 | 13.78±0.57 | **65.28±1.02** |
| | GDUMB (PRABHU ET AL., 2020) | 73.29±1.82 | 63.55±5.62 | 42.18±2.05 | 3.67±0.25 | 43.29±2.53 |
| | $k$-**FPF-CE** | 81.28±0.71 | 76.72±1.94 | 64.35±0.87 | 19.57±0.37 | 65.90±0.72 |
| | $k$-**FPF-KD** | 85.16±0.67 | **79.20±3.89** | 66.43±0.50 | **20.56±0.32** | 66.42±2.21 |
| | ER (RIEMER ET AL., 2018) | 80.45±0.99 | 57.54±3.05 | 57.64±4.27 | 10.09±0.34 | 52.72±4.01 |
| | **FPF+ER** | 84.07±1.26 | 69.83±2.87 | 65.47±2.64 | 18.61±0.70 | 64.27±1.91 |
| 500 | AGEM (CHAUDHRY ET AL., 2018) | 24.00±0.18 | 27.33±3.93 | 19.47±0.03 | 7.14±0.10 | 35.29±4.94 |
| | **FPF+AGEM** | 79.86±0.88 | 73.32±3.73 | 57.84±1.98 | 17.35±0.65 | 62.40±1.89 |
| | iCARL (REBUFFI ET AL., 2017) | 82.95±0.47 | 57.67±1.13 | 62.26±1.09 | 14.81±0.37 | - |
| | **FPF+iCARL** | 84.53±0.37 | 74.35±4.89 | 67.75±0.67 | 17.37±0.35 | - |
| | FDR (BENJAMIN ET AL., 2018) | 76.62±1.81 | 40.08±4.13 | 43.52±1.74 | 11.33±0.33 | 48.50±4.67 |
| | **FPF+FDR** | 82.32±0.91 | 75.59±2.64 | 63.82±0.69 | 17.94±0.56 | 65.47±1.13 |
| | DER (BUZZEGA ET AL., 2020) | 82.52±0.52 | 66.71±3.40 | 55.98±3.35 | 11.54±0.70 | 47.63±3.85 |
| | **FPF+DER** | 85.24±0.55 | 74.80±3.45 | 67.52±0.83 | 17.60±0.50 | 65.69±1.66 |
| | DER++ (BUZZEGA ET AL., 2020) | 84.25±0.47 | 71.09±2.60 | 67.06±0.31 | 17.14±0.66 | 57.77±2.54 |
| | **FPF+DER++** | **85.67±0.23** | 77.37±1.32 | **69.09±0.74** | 20.17±0.35 | **66.89±1.32** |

**Implementation Details.** We follow the settings in (Buzzega et al., 2020) to train various SOTA CL methods on different datasets, except training each task for only 5 epochs, which is more practical than 50 or 100 epochs in (Buzzega et al., 2020) for the streaming setting of CL. Since the epochs are reduced, we re-tune the learning rate and hyper-parameters for different scenarios by performing a grid-search on a validation set of 10% samples drawn from the original training set.

For both FPF and $k$-FPF, we use the same optimizer, i.e., SGD with the cosine-annealing learning rate schedule, and finetune the selected parameters with a batchsize of 32 for all scenarios. The finetuning steps for FPF and $k$-FPF are 300 and 100 respectively. We perform a grid-search on the validation set to tune the learning rate and other hyper-parameters. Please refer to the Appendix for the hyper-parameters we explored.

**Baseline methods.** We apply FPF to several SOTA memory-based CL methods: ER (Riemer et al., 2018), iCaRL (Rebuffi et al., 2017), A-GEM (Chaudhry et al., 2018), FDR (Benjamin et al., 2018), DER (Buzzega et al., 2020), and DER++(Buzzega et al., 2020). Besides, we also compare our methods with GDUMB (Prabhu et al., 2020) and oEWC (Schwarz et al., 2018). We report the test accuracy of these baseline methods and the best test accuracy of FPF and $k$-FPF among a few choices of finetuned parameters. We take JOINT as the upper bound for CL, which trains all tasks jointly, and SGD as the lower bound, which trains tasks sequentially without any countermeasure to forgetting. For FPF, $k$-FPF, and all memory-based methods, the performance with buffer size 200 and 500 are reported. All results reported in Table1 are averaged over five trials with different random seeds.

## 6.1 MAIN RESULTS

**FPF considerably improves the performance of all memory-based CL methods** and achieves SOTA performance over all scenarios in class-IL and domain-IL in Table 1. For methods with catastrophic forgetting, like AGEM, the accuracy of FPF increases exponentially. The surge of performance illustrates that FPF can eliminate bias by finetuning task-specific parameters to adapt to all seen tasks.

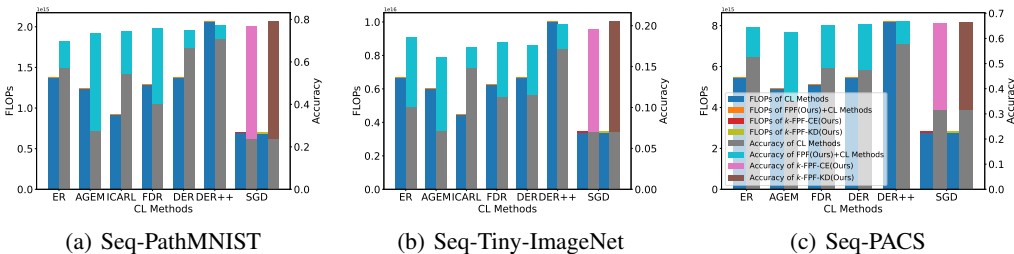

(a) Seq-PathMNIST     (b) Seq-Tiny-ImageNet     (c) Seq-PACS

Figure 4: Comparison of FLOPs and accuracy between FPF, $k$-FPF and SOTA CL methods. **FPF improves all CL methods by a large margin without notably extra computation.** $k$-FPF consumes much less computation but achieves comparable performance as FPF.

$k$**-FPF-CE replaces the costly every-step replay with efficient occasional FPF.** In Table 1, the performance of $k$-FPF-CE on Seq-PathMNIST, Seq-Tiny-ImageNet and Seq-PACS are better than the best CL methods and its performance on Seq-OrganAMNIST and Seq-Cifar10 are also better than most CL methods, which implies that finetuning the task-specific parameters on a small number of buffer during SGD can help retain the previous knowledge and mitigate forgetting, every-step replay is not necessary. In Fig. 4, the number of training FLOPs and accuracy of different methods are reported. Compared to the training FLOPs of several CL methods, the computation cost of FPF and $k$-FPF-CE is almost negligible. The overall training FLOPs of $k$-FPF-CE is still much less than SOTA CL methods while its performance are better, which show the efficiency of $k$-FPF-CE.

$k$**-FPF-KD further improves the performance of $k$-FPF-CE to be comparable to FPF.** $k$-FPF-CE propose the efficiency of CL methods, but its performance is a bit worse than that of FPF. One of the most difference between $k$-FPF and FPF is the experience replay during training of CL. Inspired by DER, we propose $k$-FPF-KD, which uses knowledge distillation to match the outputs of previous models on buffered data, hence retaining the knowledge of previous tasks. The results of $k$-FPF-KD in Table 1 show that it is comparable to FPF in most scenarios. Fig. 4 shows that the FLOPs of $k$-FPF-KD is similar to $k$-FPF-CE but much less than other CL methods and FPF, and in some cases, it outperforms FPF. $k$-FPF-KD shows SOTA performance in both efficiency and accuracy.

## 6.2 Comparison of finetuning different parameters in FPF and $k$-FPF

**FPF and $k$-FPF get the best performance when only a small portion of task-specific parameters are finetuned.** In Fig. 5, the accuracy, training FLOPs and number of trainable parameters during finetune of applying FPF or $k$-FPF to different task-specific parameters in ResNet-18 on Seq-PathMNIST are compared. Over all different scenarios, $k$-FPF only needs about half FLOPs of FPF with better performance (indicated by Red Stars). When finetuning on different task-specific parameters, FPF performs the best when BN+FC layers are finetuned, which is only $0.127\%$ of all parameters (indicated by Orange Stars). This is consistent with our observations in empirical studies where BN and FC layers are the most sensitive parameters to distribution shift. And the results shows that only finetuning a small portion of task-specific parameters can mitigate catastrophic forgetting and generalize the model.

The phenomenon for $k$-FPF is a little different. (1) In the bottom plot of Fig. 5, when FC layer is not selected for finetuning in $k$-FPF, the performance is much worse. This is because in class-IL, the output classes change across tasks so the FC layer is trained to only output the classes for the current task (Hou et al., 2019). In contrast, when applying $k$-FPF to domain-IL on Seq-PACS, where the output classes keep the same for different tasks, Fig. 11 in Appendix shows that finetuning FC layer performs similarly as finetuning other parameters. Hence, the last FC layer is more sensitive in class-IL than in Domain-IL. This is also shown Fig. 2 (a), (d). (2) As the red star indicates, $k$-FPF needs to finetune a little more parameters (Block3 of convolutional layers, $18.91\%$ of all parameters) to achieve a comparable accuracy with FPF. Without experience replay during SGD, the model has a larger bias on the current task and thus more task-specific parameters are needed to be finetuned. This also indicates that such bias of task-specific parameters is the main reason for catastrophic forgetting. When Block4 ($75.22\%$ of all parameters) is finetuned, since it is the most stable parameters in our empirical study, the performance of $k$-FPF degrades.

## 6.3 Analysis of FPF and $k$-FPF in Different Scenarios

**Different training FLOPs for**
$k$**-FPF** In Fig. 6(a), we study the
trade-off between the training
FLOPs and the accuracy of
$k$-FPF on Seq-PathMNIST by
changing $k$ and the number
of finetuning steps. $\tau$ in the
legend refers to the interval of
two consecutive FPF. Fixing $k$,
$k$-FPF saturates quickly as the
finetuning steps increase. This
implies that $k$-FPF is efficient
on FLOPs to achieve the best
performance. For experiments
with small $k$, e.g. $k$=2, though
the computation required is very
low, performance cannot be
further improved. This implies
that FPF needs to be applied
on buffered samples more fre-
quently to mitigate forgetting.

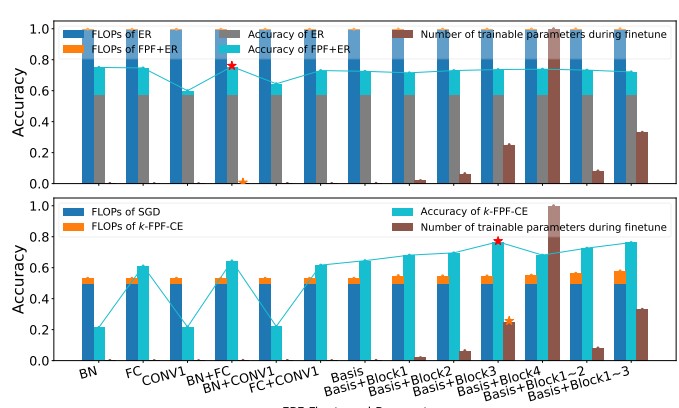

Figure 5: Comparison of FLOPs, number of finetuned parameters, and
accuracy for FPF(Top) and $k$-FPF(Bottom) finetuning different combi-
nations of parameters. All FLOPs are normalized together to (0,1], as
well as the number of finetuning parameters. "Basis" in the x-label refers
to "BN+FC+CONV1". Red stars highlight the best accuracy and show
**both FPF and $k$-FPF only require to finetune a small portion of task-
specific parameters. $k$-FPF halves FPF's FLOPs.**

When $k$ is large, e.g., $k$=41 or 121, the accuracy slightly improves with the price of much more
required computation. As the red star in the plot indicates, applying FPF every 1500 training steps
can achieve the best computation-accuracy trade-off.

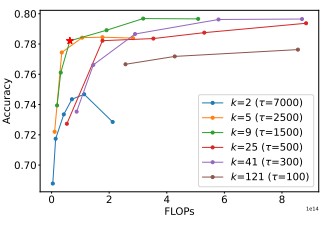

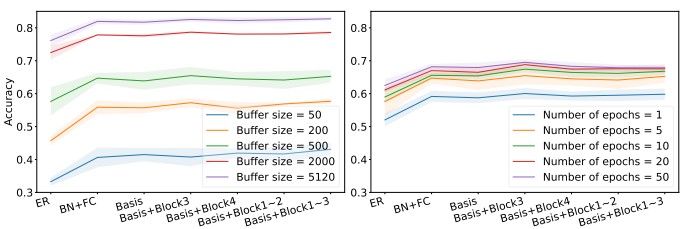

(a) FLOPs-Accuracy in $k$-FPF          (b) Different buffer sizes and training epochs for FPF

Figure 6: **(a)** Trade-off between FLOPs and accuracy for $k$-FPF with different $k$ and $\tau$ (the SGD steps between
two consecutive FPF). By increasing the finetunine steps per FPF, the accuracy quickly saturates. The best
trade-off is highlighted at the top-left corner when $k = 9(\tau = 1500)$. **(b)** Comparison between ER and
FPF+ER finetuning different parameters with different buffer sizes and number of epochs per task. In all
scenarios, FPF can significantly improve ER by only finetuning BN+FC.

**Different buffer sizes and training epochs for FPF** The buffer size and the training epochs
per task are usually crucial in memory-based CL methods. In Fig. 6(b), when the buffer size or
number of epochs increases, the performance of ER improves as well. However, increasing the
buffer size brings more benefits. When the buffer size or epochs grow too large, the performance
of ER seems saturate and increases slowly. For all scenarios, finetuning BN+FC layers are highly
effective to alleviate the current task's bias and promote the performance, which is consistent with
our observations from the empirical studies.

# 7    CONCLUSION

We study a fundamental problem in CL, i.e., which parts of a neural network are task-specific and
more prone to catastrophic forgetting. Extensive empirical studies in diverse settings consistently
show that only a small portion of parameters is task-specific and sensitive. This discovery leads to
a simple yet effective "forgetting prioritized finetuning (FPF)" that only finetunes a subset of these
parameters on the buffered data before model deployment. FPF is complementary to existing CL
methods and can consistently improve their performance. We further replace the costly every-step
replay with $k$-times of occasional FPF during CL to improve the efficiency. Such $k$-FPF achieves
comparable performance as FPF+SOTA CL while consumes nearly half of its computation. In
future work, we will study how to further reduce the memory size required by FPF.

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

# A APPENDIX

## A.1 SELECTION OF THE SENSITIVE PART OF DIVERSE TYPES OF NEURAL NETWORKS

We select sensitive parameters according to their training dynamics in the early epochs. Examples of the training dynamics for different layers are shown in Fig 1-2 and their ranking does not change over epochs. Specifically, we sort layers by their training dynamics values in descent order. Then we greedily add layers one after another to the sensitive group until the sum of all selected layers' training dynamics exceeds $p$ percent of the sum for all layers, where $p$ is a hyper-parameter. In our experiments, for models with the batch-norm layer like ResNet-18, FPF and $k$-FPF outperforms all baselines when $p = 97$, when only $18.90\%$ of parameters in the neural network are regarded as sensitive parameters. For other models like MLP and VGG-11, $p = 70$ and only $1.12\%$ and $0.32\%$ of parameters are regarded as sensitive parameters.

## A.2 COMPARISON BETWEEN FPF AND THE METHOD FINE-TUNING ALL PARAMETERS

In Tab.2, we compare FPF with FPF-ALL (which finetunes all parameters) when applied to different CL methods for two types of CL, i.e., class-IL and domain-IL. The results shows that FPF consistently achieve comparable or slightly higher accuracy than FPF-ALL by spending significantly less FLOPs. This demonstrates the advantage of FPF on efficiency.

Table 2: Comparison of accuracy and FLOPs between FPF and FPF-ALL(finetuning all parameters).

| Methods | Seq-PathMNIST | | Seq-PACS | |
|---|---|---|---|---|
| | Accuracy | FLOPs(B) | Accuracy | FLOPs(B) |
| $k$-**FPF-CE** | 76.72±1.94 | 21.35 | 65.90±0.72 | 148.25 |
| $k$-**FPF-ALL-CE** | 75.74±2.91 | 43.95 | 64.48±2.23 | 174.60 |
| **FPF**+ER | 69.83±2.87 | 4.68 | 64.27±1.91 | 24.39 |
| **FPF-ALL**+ER | 70.64±4.00 | 8.79 | 63.81±2.33 | 34.92 |
| **FPF**+AGEM | 73.32±3.73 | 7.07 | 62.40±1.89 | 18.47 |
| **FPF-ALL**+AGEM | 74.80±3.12 | 8.79 | 62.65±1.65 | 34.92 |
| **FPF**+iCaRL | 74.35±4.89 | 4.27 | - | - |
| **FPF-ALL**+iCaRL | 72.77±4.12 | 8.79 | - | - |
| **FPF**+FDR | 75.59±2.64 | 2.94 | 65.47±1.13 | 11.70 |
| **FPF-ALL**+FDR | 74.24±1.48 | 8.79 | 64.88±2.28 | 34.92 |
| **FPF**+DER | 74.80±3.45 | 2.96 | 65.69±1.66 | 18.47 |
| **FPF-ALL**+DER | 74.54±3.19 | 8.79 | 66.22±0.87 | 34.92 |
| **FPF**+DER++ | 77.37±1.32 | 4.68 | 66.89±1.32 | 24.39 |
| **FPF-ALL**+DER++ | 77.16±1.45 | 8.79 | 65.19±1.33 | 34.92 |

## A.3 EXPERIMENTS ON THE TASK SEQUENCE CONTAINING TOTALLY DIFFERENT DATASETS

We concatenate the CL tasks from two datasets (i.e., Seq-CIFAR-10 and Seq-PathMNIST) to form a twice-longer task sequence and evaluate the training dynamics of different groups of parameters. The result is shown in Fig.7. The vertical dashed line at epochs $= 30$ is the boundary between the two datasets. Although the two datasets are from different domains and thus have different distributions, the sensitivity of parameters stays consistent with our observations on tasks from a single dataset. This indicates that the sensitive parameters almost do not change across different tasks/datasets so they can be identified at very early stages.

## A.4 A MORE CLEAR VERSION OF FIG. 4 AND FIG.5

In Fig.8 and Fig.9, to make Fig.4 and Fig.5 more concise and easy to understand, we draw the barplots of different parts separately.

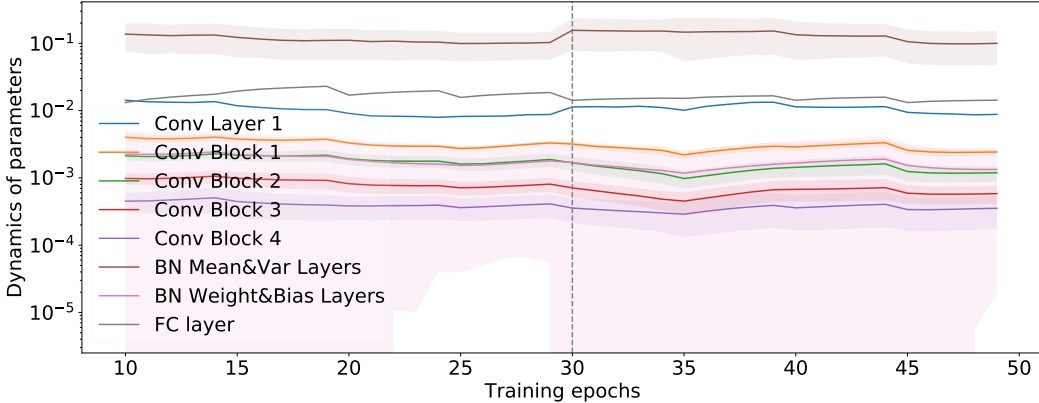

Figure 7: The training dynamics of different groups of parameters in ResNet-18 when sequentially training Seq-CIFAR-10 and Seq-PathMNIST.

### A.5 PERFORMANCE OF VARIOUS METHODS DURING THE TRAINING OF CL

In Tab. 3 and Tab. 4, the average test accuracy of previous tasks at the end of each task during the training of CL on Seq-PathMNIST and Seq-PACS is reported. The results show that during training, $k$-FPF can always achieve the best performance among various CL methods. Whenever the training stops, $k$-FPF can always achieve a model performing well on previous tasks.

Table 3: The average accuracy of previous tasks at the end of each task during the training of CL on Seq-PathMNIST.

| Methods | Task 1 | Task 2 | Task 3 | Task 4 |
|---------|--------|--------|--------|--------|
| $k$-FPF-CE | 99.95±0.04 | 95.41±1.98 | 81.92±2.26 | 76.72±1.94 |
| ER | 98.62±1.59 | 83.06±3.12 | 74.60±3.18 | 57.54±3.05 |
| AGEM | 99.71±0.19 | 46.58±3.13 | 36.12±3.17 | 27.33±3.93 |
| iCaRL | 99.98±0.02 | 86.86±5.47 | 66.62±5.64 | 57.67±1.13 |
| FDR | 99.97±0.06 | 48.06±0.82 | 55.75±6.55 | 40.08±4.13 |
| DER | 99.98±0.02 | 91.92±3.42 | 76.50±5.77 | 66.71±3.40 |
| DER++ | 99.95±0.06 | 94.06±6.14 | 80.35±3.32 | 71.09±2.60 |

Table 4: The average accuracy of previous tasks at the end of each task during the training of CL on Seq-PACS.

| Methods | Task 1 | Task 2 | Task 3 | Task 4 |
|---------|--------|--------|--------|--------|
| $k$-FPF-CE | 70.94±2.02 | 73.75±2.68 | 62.37±0.49 | 65.90±0.72 |
| ER | 56.64±9.04 | 54.34±9.44 | 46.79±8.48 | 52.72±4.01 |
| AGEM | 47.34±7.35 | 38.02±5.82 | 32.70±7.13 | 35.29±4.94 |
| FDR | 58.59±4.36 | 54.00±4.01 | 46.38±4.80 | 48.50±4.67 |
| DER | 48.49±9.40 | 45.28±8.88 | 34.48±7.81 | 47.63±3.85 |
| DER++ | 55.33±7.45 | 64.43±6.50 | 50.19±7.30 | 57.77±2.54 |

### A.6 DETAILED DYNAMICS OF BN WEIGHTS AND BIAS IN DIFFERENT GROUPS

In Fig. 10, the training dynamics of BN weights and biases in different groups are reported. This provides a fine-grained explanation to the phenomenon in Fig. 1 (c): the bottom BN layer is much more sensitive and task-specific than other BN layers. Consistent with convolutional layers, the deep BN layers are less sensitive to task drift than the shallower ones.

In a neural network, lower layers are closer to the input. Since the distribution of the inputs changes, the parameters of lower convolutional layers change sensitively to adapt to the distribution shift. The weights and biases of BN, which are the scale and shift of the featuremap, will change along

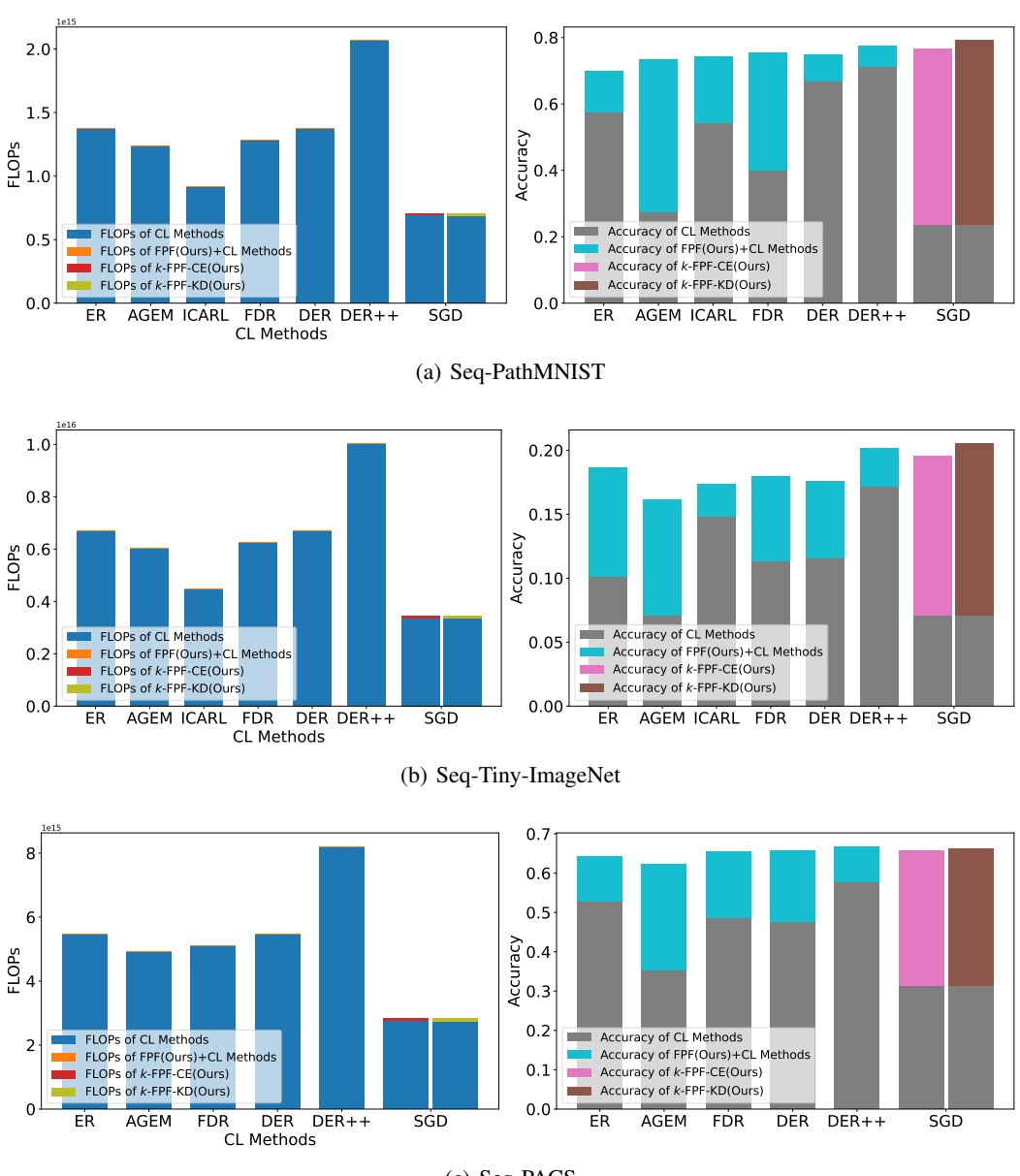

(a) Seq-PathMNIST

(b) Seq-Tiny-ImageNet

(c) Seq-PACS

Figure 8: Comparison of FLOPs and accuracy between FPF, $k$-FPF and SOTA CL methods. **FPF improves all CL methods by a large margin without notably extra computation.** $k$-**FPF consumes much less computation but achieves comparable performance as FPF.**

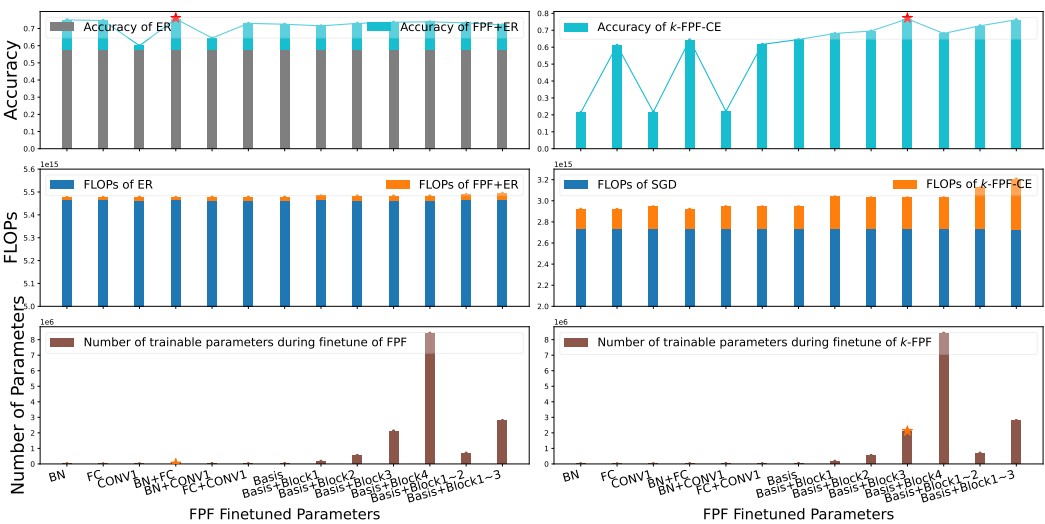

Figure 9: Comparison of FLOPs, number of finetuned parameters, and accuracy for FPF(Top) and $k$-FPF(Bottom) finetuning different combinations of parameters. All FLOPs are normalized together to (0,1], as well as the number of finetuning parameters. "Basis" in the x-label refers to "BN+FC+CONV1". Red stars highlight the best accuracy and show **both FPF and $k$-FPF only require to finetune a small portion of task-specific parameters. $k$-FPF halves FPF's FLOPs.**

with the convolutional parameters to adjust the distribution of the output featuremap. In the deeper layers, the functionality of each filter is relatively stable, so the distribution of the featuremap need not change drastically.

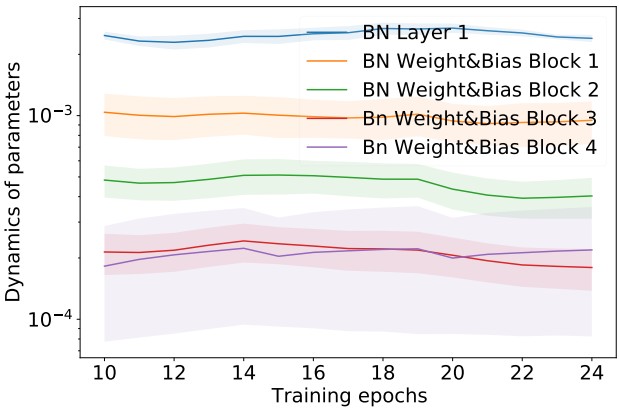

Figure 10: The training dynamics of different groups of BN weights and biases in ResNet-18.

## A.7 RESULTS OF OTHER NEURAL NETWORKS

In the Tab. 5, the results of various CL benchmarks and FPF on MLP and VGG-11 are reported. Similar to the results in Tab.1, by finetuning the most sensitive parameters in MLP and VGG-11, FPF can further improve the performance of all SOTA CL methods and achieve the best performance. $k$-FPF-CE also achieves comparable performance as FPF + SOTA methods. Our methods can generalize to various neural networks.

Table 5: Classification results for CL benchmarks and FPF on MLP and VGG-11. **Bold** and underline indicate the best and second best algorithms in each setting.

| BUFFER | METHODS | CLASS-IL | |
| --- | --- | --- | --- |
| | | SEQ-MNIST(MLP) | SEQ-CIFAR10(VGG-11) |
| | JOINT | 95.58±0.33 | 69.50±0.73 |
| | SGD | 19.64±0.07 | 18.71±0.33 |
| | OEWC | 20.69±1.34 | 18.46±0.23 |
| | GDUMB | 90.60±0.37 | 41.65±0.78 |
| | $k$-FPF-CE | 90.63±0.57 | 55.45±1.16 |
| | ER | 86.73±1.03 | 46.27±1.18 |
| | FPF+ER | 91.15±0.16 | 53.48±1.08 |
| | AGEM | 51.03±4.94 | 19.40±1.09 |
| | FPF+AGEM | 89.26±0.52 | 29.84±1.37 |
| 500 | ICARL | 58.12±1.94 | 45.63±1.94 |
| | FTF+ICARL | 80.83±0.49 | 48.03±0.65 |
| | FDR | 83.79±4.15 | 45.56±2.23 |
| | FPF+FDR | 89.67±0.37 | 55.59±1.56 |
| | DER | 91.17±0.94 | 51.12±2.47 |
| | FPF+DER | **91.25±0.89** | **57,46±1.15** |
| | DER++ | 91.18±0.74 | 47.60±3.23 |
| | FTF+DER++ | 91.22±0.67 | 54.69±0.73 |

## A.8 PERFORMANCE OF FINETUNING DIFFERENT PARAMETERS FOR FPF AND K-FPF ON DOMAIN-IL DATASET

In Figure 11, the performance of finetuning different parameters for FPF and $k$-FPF on domain-IL dataset Seq-PACS are reported.

## A.9 COMPARISON WITH RELATED WORKS (RAMASESH ET AL., 2020)

Paper "Anatomy of catastrophic forgetting: Hidden representations and task sementics" shows that freezing bottom layers had little impact on the performance of the second task. (i) Their setting is different: our study and most CL methods focus on the performance of ALL tasks. And it is unfair in terms of parameter amount to compare freezing effects of multiple layers/blocks (e.g., block 1-3) vs. one layer/block. (ii) Their result is partially consistent with ours since their unfrozen part covers the last layer and many BN parameters, which are the most sensitive/critical part to finetune in our paper. (iii) The rest difference is due to our finer-grained study on parameters and on $> 2$ tasks but this paper only studies two tasks and focuses on the second. Table 6 shows the class-IL accuracy at the end of each task if freezing different single ResNet block (bottom to top: block-1 to block-4). At the end of task-2, our observation is the same as this paper and freezing bottom blocks showing little reduction of accuracy. However, at the end of task 3-5, their performance drops and freezing block-1 drops most.

## A.10 HYPER-PARAMETER SEARCH SPACE

In the following, we provide a list of all the hyper-parameter combinations that were considered for FPF and $k$-FPF.

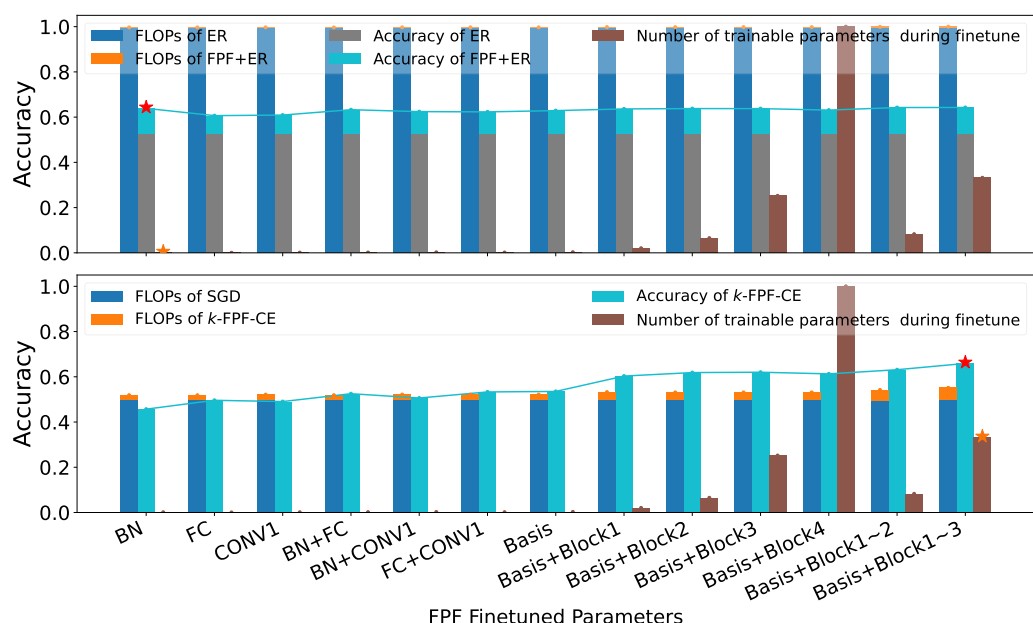

Figure 11: Comparison of FLOPs, number of finetuned parameters, and accuracy for FPF(Top) and $k$-FPF(Bottom) finetuning different combinations of parameters. All FLOPs are normalized together to (0,1], as well as the number of finetuning parameters. "Basis" in the x-label refers to "BN+FC+CONV1". Red stars highlight the best accuracy and show both FPF and $k$-FPF only require to finetune a small portion of task-specific parameters. $k$-FPF halves FPF's FLOPs. **Different from the results of $k$-FPF in class-IL, in Seq-PACS, since the output classes for different tasks are always the same, the last FC layer will not have a large bias on particular classes. Only finetuning BN or CONV1 layers for $k$-FPF can get comparable performance with ER.** Similar to class-IL, since experience replay is not allowed during the training of CL method SGD, a little more parameters are required to be finetuned by $k$-FPF to get comparable performance with FPF (about $24.92\%$ of all parameters).

Table 6: class-IL accuracy of ER at the end of each task on Seq-CIFAR-10

|  | Task-1 | Task-2 | Task-3 | Task-4 | Task-5 |
|---|---|---|---|---|---|
| No Freeze | $97.52 \pm 0.23$ | $80.53 \pm 0.80$ | $63.96 \pm 0.51$ | $58.05 \pm 1.91$ | $57.03 \pm 2.29$ |
| Freeze conv-1 | $97.52 \pm 0.23$ | $79.62 \pm 2.75$ | $63.28 \pm 2.13$ | $56.11 \pm 0.61$ | $55.58 \pm 1.31$ |
| Freeze block-1 | $97.52 \pm 0.23$ | $78.88 \pm 3.01$ | $60.07 \pm 0.61$ | $55.49 \pm 0.22$ | $52.75 \pm 1.90$ |
| Freeze block-2 | $97.52 \pm 0.23$ | $78.93 \pm 3.34$ | $63.78 \pm 2.32$ | $56.23 \pm 0.82$ | $56.55 \pm 3.17$ |
| Freeze block-3 | $97.52 \pm 0.23$ | $80.37 \pm 2.35$ | $64.31 \pm 2.23$ | $57.21 \pm 0.40$ | $56.52 \pm 0.76$ |
| Freeze block-4 | $97.52 \pm 0.23$ | $80.68 \pm 1.53$ | $64.89 \pm 1.00$ | $53.78 \pm 3.37$ | $54.01 \pm 2.07$ |

Table 7: The hyper-parameter search space for FPF on different datasets. For all experiments of FPF, we use the same number of batch size 32 and finetuning steps 300. The hyper-parameter spaces of finetuning different parameters in the models generated by different CL methods are always same for a given dataset. ft-lr refers to the learning rate during finetuning of FPF.

| Dataset | Hyper-parameter | Values |
|---|---|---|
| Seq-OrganAMNIST | lr | [1, 0.3, 0.1, 0.03, 0.01] |
| Seq-PathMNIST | lr | [1, 0.75, 0.3, 0.05, 0.03] |
| Seq-CIFAR-10 | lr | [1, 0.3, 0.1, 0.03, 0.01] |
| Seq-Tiny-ImageNet | lr | [1, 0.5, 0.3, 0.075, 0.05] |
| Seq-PACS | lr | [1, 0.5, 0.3, 0.05, 0.03, 0.005, 0.003] |

Table 8: The hyper-parameter search space for $k$-FPF-SGD on different datasets. For all experiments of $k$-FPF-SGD, we use the same number of batch size 32 and finetuning steps 100. The hyper-parameter spaces of finetuning different parameters are always same for a given dataset. lr refers to the learning rate during training of CL method SGD. ft-lr refers to the learning rate during finetuning.

| Dataset | Hyper-parameter | Values |
|---|---|---|
| Seq-OrganAMNIST | lr | [0.2, 0.15, 0.1, 0.075] |
|  | ft-lr | [0.5, 0.2, 0.15, 0.1] |
| Seq-PathMNIST | lr | [0.05, 0.03, 0.01] |
|  | lr | [0.1, 0.075, 0.05, 0.03, 0.01] |
| Seq-CIFAR-10 | lr | [0.05, 0.03, 0.01] |
|  | ft-lr | [0.075, 0.05, 0.03, 0.01] |
| Seq-Tiny-ImageNet | lr | [0.075, 0.05, 0.03] |
|  | ft-lr | [0.1, 0.075, 0.05] |
| Seq-PACS | lr | [0.05, 0.03, 0.01] |
|  | ft-lr | [0.075, 0.05, 0.03, 0.0075] |

Table 9: The hyper-parameter search space for $k$-FPF-KD on different datasets. For all experiments of $k$-FPF-KD, we use the same number of batch size 32 and finetuning steps 100. The hyper-parameter spaces of finetuning different parameters are always same for a given dataset. lr refers to the learning rate during training of CL method SGD. ft-lr refers to the learning rate during finetuning. $\lambda$ is the hyper-parameter to balance the two losses.

| Dataset | Hyper-parameter | Values |
|---|---|---|
| Seq-OrganAMNIST | lr | [0.2, 0.15, 0.1, 0.075] |
| | ft-lr | [0.5, 0.2, 0.15, 0.1] |
| | $\lambda$ | [1, 0.5, 0.2, 0.1] |
| Seq-PathMNIST | lr | [0.05, 0.03, 0.01] |
| | lr | [0.1, 0.075, 0.05, 0.03, 0.01] |
| | $\lambda$ | [1, 0.5, 0.2, 0.1] |
| Seq-CIFAR-10 | lr | [0.05, 0.03, 0.01] |
| | ft-lr | [0.075, 0.05, 0.03, 0.01] |
| | $\lambda$ | [0.5, 0.2, 0.1] |
| Seq-Tiny-ImageNet | lr | [0.075, 0.05, 0.03]] |
| | ft-lr | [0.1, 0.075, 0.05] |
| | $\lambda$ | [1, 0.5, 0.2] |
| Seq-PACS | lr | [0.05, 0.03, 0.01] |
| | ft-lr | [0.075, 0.05, 0.03, 0.0075] |
| | $\lambda$ | [1, 0.5 0.2 0.1] |

