# OpenReview forum: "Does Continual Learning Equally Forget All Parameters?"
_ICLR.cc/2023/Conference — Submitted to ICLR 2023_

### Official Review · Reviewer_pB6S · 2022-10-19

**Confidence:** 3
**Correctness:** 3
**Technical Novelty And Significance:** 3
**Empirical Novelty And Significance:** 3
**Recommendation:** 6

**Clarity, Quality, Novelty And Reproducibility:**

Clarity is good.
Quality is overall good but there are a few issues that need to be resolved.
Novelty is fair.
Reproducibility: the authors are expected to release the source code to the public.

**Strength And Weaknesses:**

Upsides:
+ This paper analyzes the sensitivity of different network parameters in continual learning, and shows some interesting observations.
+ The empirical studies are solid and the writing is easy to follow.
+ A new Forgetting Prioritized Fine-tuning strategy is proposed. By lazily conducting replay, the proposed method achieves comparable performance with lower computation costs.


Downsides:
- How to adaptively select the task-specific parameters for fine-tuning? This paper only provides an empirical study for it, but a well-designed adaptive strategy is expected based on your empirical studies. In this case, the contribution of this paper will become more significant.
- In Table 1, does any specific method perform the best with the proposed FPF? It seems that there is no consistently state-of-the-art method when combining FPF.
- The colors of Figure 4&5 are too complex, making it hard to understand. It would be better to simplify this figure.
- Many previous studies find that the higher layers are more sensitive. Please discuss more Figure 7. Why are lower BN layers more sensitive？ There may be some interesting discussions.

**Summary Of The Paper:**

This paper conducts a series of empirical studies for continual learning about the sensitivity of different network parameters, and shows some interesting observations. Based on the observations, this paper explores a Forgetting Prioritized Fine-tuning strategy for continual learning. Empirical results demonstrate its effectiveness.

**Summary Of The Review:**

Overall, I like the empirical observation of this paper which may inspire some future research. However, there are still some questions to be resolved. I expect to responses from the authors.

**********Post rebuttal*************
After discussion with the authors and other reviewers, I suggest the authors to improve the paper by further analyzing the found observations about their in-depth reasons and making the proposed method more adaptive regarding the selection of layers and hyper-paramters.

---

> ### Author Response · Authors · 2022-11-17
> **Response to Reviewer pB6S**
>
> Thanks for your comments and your efforts for reviewing our paper! In the following, we provide answers to your questions.
>
> **1. How to adaptively select the task-specific parameters for fine-tuning?**
>
> Please refer to our reply to Question 1 in Response to all reviewers.
>
> **2. Does any specific method perform the best with the proposed FPF?**
>
> In Tab.1 of the paper, the results show that FPF can always significantly improve the performance of existing CL methods.
> Although no specific method can always achieve the best performance in all settings, FPF+DER++ and $k$-FPF-KD achieve the best or second-best performance in more than $90$% settings. If the computation is limited, $k$-FPF-KD is the best choice. For better accuracy, FPF+DER++ is more preferred.
>
> **3. The colors of Figure 4&5 are too complex.**
>
> Thank you for your suggestion! For a better comparison between different methods and due to the space limit, we report the FLOPs and Accuracy in the same plot. In Fig.8 and Fig.9 of the Appendix in the revision, we report the FLOPs and Accuracy separately.
>
> **4. Why are lower BN layers more sensitive?**
>
> Our observations of sensitivity are consistent on both BN weights & biases from different blocks and convolutional layers, i.e., the deep layers are less sensitive to the distribution drift of CL than the shallower ones.
>
> In a neural network, lower layers are closer to the input. Since the distribution of the inputs changes, the parameters of lower convolutional layers change sensitively to adapt to the distribution shift. The weights and biases of BN, which are the scale and shift of the featuremaps, will change along with the convolutional parameters to adjust the distribution of the output featuremap.
> In the deeper layers, the functionality of each filter is relatively stable, so the distribution of the featuremap need not change drastically.

---

> > ### Comment · Reviewer_pB6S · 2022-11-18
> > **Thanks for the response.**
> >
> > Thanks for the response. It addresses most of my concerns. The selection method is still empirical and not that adaptive, so I keep the original score of 6.

---

> > > ### Author Response · Authors · 2022-12-01
> > > **Thank you for your review!**
> > >
> > > We are glad to hear from you that most of your concerns have been successfully addressed by our rebuttal.
> > >
> > > The selection of sensitive parameters is merely one of several original contributions of our paper.
> > > Besides, in our reply to point 4, our method can be used to compare the sensitivities of different parameters from a global perspective.
> > > Moreover, as we have replied to point 2, our proposed FPF and $k$-FPF considerably outperform SOTA continual learning methods on both accuracy and efficiency by a large margin.
> > >
> > > We would like to thank you again for your thorough review and insightful suggestions. We believe they can improve the overall quality of our paper. We sincerely encourage the reviewer to revise the awarded rating based on our updated version, new results,  and clarifications.

---

### Official Review · Reviewer_8gaT · 2022-10-23

**Confidence:** 5
**Correctness:** 2
**Technical Novelty And Significance:** 2
**Empirical Novelty And Significance:** 2
**Recommendation:** 3

**Clarity, Quality, Novelty And Reproducibility:**

As anticipated in the assessment above:
 - Reproducibility: weak due to the absence of details about the implementation of the FPF strategies
 - Novelty: weak due to the proposed approach being based on an unsurprising rule-of-thumb implemented by a metric (instantaneous parameter change between tasks) which provides little novel insight into the problem and whose application hardly generalizes over simple CL scenarios.
 - Clarity and quality: weak, due to linguistic issue and lack of details about the FPF.


**Strength And Weaknesses:**

-- STRENGTHS –
(+) The empirical analysis is quite articulated both in terms of number of benchmarks being considered as well as in terms of related CL strategies being assessed. The work is mostly empirical, both in its conception and its execution, so this part is quite well-done.

(+) The generality of the approach is certainly positive: the fact that it can be applied to existing CL strategies is a plus, which stems from the simplicity of the approach.

(+) A careful consideration for the computational costs, in a research topic that is often motivated by the necessity of making neural training sustainable, is an added strength and well motivates a part of the work.

-- WEAKNESSES –

(-) Despite the attempt of rooting the work on considerations about task interference and drifts, this paper is mostly based on an rule-of-thumb intuition (quite an unsurprising one) which is not backed up by convincing theoretical nor empirical motivations. The attempt at providing an empirical motivation in Section 4 is unconvincing because of there is no logical consequence between the  results of the experiments and the implementation of the FPF strategy. The hints from the empirical analysis are either trivial (i.e. fine tune only the closer-to-output-layer weights) or very architecture specific (i.e. in some cases it is best to tune the outer layers but also the early layers, but not the middle ones). This kind of insight is not actionable by the community as it is not enough general.

(-) Generality and impact of the strategy are also severely limited by the choice of the CL scenarios. It is straightforward to see how a metric based on instantaneous change in weights between adjacent tasks can work in CI and DI settings, as in these cases there is a clear distinction between tasks at training time, which in the CI setting is also mirrored in task-specific components of the architecture (multi-heads). The work could gain considerable strength if it can be shown that the insights and strategies can work less ideal scenarios, such as class incremental with repetition (reoccurring tasks) and single class incremental learning (no architectural segregation between tasks).

(-) The exact details of the FPF strategies are nowhere to be found. Only a vague description of the intuition is provided by the details of the strategies are not provided. For instance it is not clear (until one reaches the experiments) if a buffer is needed, and what is the strategy to populate it. One would like to know how one chooses the portions of the network to include/exclude in FPF. By reading the paper it should be possible to replicate the strategy on different problems and architectures, but this is not the case given the lack of details in the current formulation. The paper should definitely be extend to include pseudo-code and other formal description of the strategies to make everything reproducible and generalizable.

(-) Linguistic quality is sometimes borderline, especially in parts of the experimental section whose writing might had been rushed. Another round of proofreading is highly advised.


**Summary Of The Paper:**

The paper goal is to study which part of specific neural architectures is mostly affected by catastrophic forgetting in Class Incremental (CI) and Domain Incremental (DI) continual learning, to apply specialized fine tuning to those parts with the purpose of reducing interference. The 1-norm of $l$-th layer weight change between consecutive CL tasks is used as metric to measure which layers are affected most. Two finetuning procedures are provided, with one designed to optimize computational cost of memory consolidation. The method can be hot-plugged to several CL methods in literature.

**Summary Of The Review:**

While the paper has good motivations in abstract (finding a general and efficient procedure for parameter consolidation), it fails to deliver convincing insights, due to lack of technical detail, reduced novelty and scarce generality of the results. Overall the weaknesses quite heavily surpass the merit.

---

> ### Author Response · Authors · 2022-11-17
> **Response to Reviewer 8gaT**
>
> Thanks for your comments and your efforts for reviewing our paper! In the following, we provide answers to your questions.
>
> **1. The attempt at providing an empirical motivation in Section 4 is unconvincing because of there is no logical consequence between the results of the experiments and the implementation of the FPF strategy.**
>
> **We respectfully disagree because the empirical study directly motivates FPF and $k$-FPF**: the empirical study in Section 4 shows that only a few sensitive parameters are impacted by the distribution shift in CL, so in FPF and $k$-FPF, we only finetune these sensitive parameters using the limited buffer, which brings significant improvement on both the efficiency and the final accuracy.
>
> **2. The hints from the empirical analysis are either trivial or very architecture specific. This kind of insight is not actionable by the community as it is not enough general.**
>
> **This is a misinterpretation of our work**. Our empirical study shows that the most sensitive parameters in CL may vary across different architectures and early-stage training dynamics suffice to identify these parameters. Therefore, instead of fixing the sensitive parameters to finetune, we can use the novel metric of training dynamics to select sensitive parameters for different neural nets adaptively.
>
> **3. Generality and impact of the strategy are also severely limited by the choice of the CL scenarios.
> It is straightforward to see this metric can work in CI and DI settings. There is a clear distinction between tasks, which in the CI setting is also mirrored in task-specific components of the architecture (multi-heads)**
>
> **This is, again, a misinterpretation of our work.** We followed the common setting adopted by most previous CL papers so we could have a fair comparison with them. Specifically, in both CI and DI, different tasks share exactly the same parameters (including the final layer which outputs the probabilities for all possible classes), and there are NO task-specific components of the architecture such as multi-heads.
>
> **4. The work could gain considerable strength if it can be shown that the insights and strategies can work less ideal scenarios, such as class incremental with repetition (reoccurring tasks) and single class incremental learning (no architectural segregation between tasks.**
>
> We appreciate the reviewer for being constructive in the comments! However, reoccurring the same task is actually ''more ideal'' and easier than the setting we adopted since a model forgetting a previous task gets a second chance to restore the same task when repeating it. Moreover, we did not use ''any architectural aggregation between tasks,'' and the single class incremental setting is a non-standard setting for CL. This paper follows the most widely used setting in previous CL literature.
>
> **5. The exact details of the FPF strategies are nowhere to be found.**
>
> The Reviewer hdv9 thinks that we have provided details of our experimental setup and hyper-parameters used to reproduce the results in our paper.
>
> In Section 5 of the paper, we describe our proposed FPF and $k$-FPF in detail and compare them with SGD and replay-based methods in Fig.3 so that readers can better understand our methods.
> In Section 6, we provide implementation details of the experiments. We follow most settings of DER[1], including reserving a buffer by reservoir sampling. In Appendix, the search space of different hyper-parameters is listed.
>
> **6. One would like to know how one chooses the portions of the network to include/exclude in FPF.**
>
> Please refer to our reply to Question 1 in Response to all reviewers.
>
>
> [1] Pietro Buzzega, Matteo Boschini, Angelo Porrello, Davide Abati, and Simone Calderara. Dark experience for general continual learning: a strong, simple baseline.

---

> > ### Comment · Reviewer_8gaT · 2022-11-17
> > **Post rebuttal**
> >
> > Thank you for your response. Some of my concerns are now clearer but others aren't really.
> >
> > The claimed novelty in being the first paper in studying the sensitivity of parameters in CL is not entirely supported to the extent of my knowledge.   Most of the empirical findings of this experiment are known from previous works. For instance, the sensitivity of final layers is known through several works, but certainly this one measured it:
> > Lesort, T., George, T. & Rish, I. Continual Learning in Deep Networks: an Analysis of the Last Layer. arXiv (2021).
> > Also the sensitivity of batch norm layers has been noted a by:
> > Lomonaco, V., Maltoni, D. & Pellegrini, L. Rehearsal-Free Continual Learning over Small Non-I.I.D. Batches. in CVPR Workshop on Continual Learning for Computer Vision 246–247 (2020).
> >
> > This leaves us with the influence of early layers, which is somehow new but also quite peculiar to specific settings and architectures. I give it to the Authors when it is stated that this work is attempting at finding specialized strategies for specific models thanks to the training dynamics measure, but:
> > 1) The paper only considers an MLP and two instances of a CNN. If the intent is to focus on how the method can generalize and provide specialized insights across multiple neural model families I would expect to see a larger choice of models being assessed
> > 2) The hyperparameter p mentioned in the general response seems to vary much depending on the architectures. Then it becomes critical to be sure that such an hyperparameter can be set within the  early-stage rather than at model selection. The whole strategy as described in the rebuttal hinges on being able to do an early-stage assessment of the training dynamics to control forgetting. But if that early stage assessment needs an hyperparameter chosen by looking at the aggregated performance at the end of the training experiences, then it is not much early -stage.
> >
> > About the implementation of the hyperparameter search strategy, I would expect that the different implementation w.r.t. the setting in Buzzega et al to lead to higher accuracies (given the more scenario specific model selection strategy). Instead I see that DER++ results are lower: do you have any hint of why this happens as the network backbone seems the same.
> >
> > About this comment:
> > >Specifically, in both CI and DI, different tasks share exactly the same parameters (including the final layer which outputs the probabilities for all possible classes), and there are NO task-specific components of the architecture such as multi-heads."
> > I appreciate the absence of multi-heads. It is though not clear to me how the final layer does not have class-specific outputs dynamically addeded in the CI setting. This does not sound like the "standard CI setting".
> >
> > Finally, it would be helpful if the Authors can provide any measurement/hint has concerns what happens to the task accuracy during the stream of experiences. What I find now in the paper shows that in the end the model does not forget, but the objective is to keep the model being able to perform the learned tasks throughout the whole time. It would be good to see if this no-forgetting property holds nicely throughout the stream.

---

> > > ### Author Response · Authors · 2022-11-20
> > > **New Response to Reviewer 8gaT (2)**
> > >
> > >
> > > **5. It is though not clear to me how the final layer does not have class-specific outputs dynamically added in the CI setting. This does not sound like the "standard CI setting".**
> > >
> > > We follow the standard setting used almost by all recent continual learning works for class-IL (we list a few papers for examples below). In our implementation, we use exactly the same code of DE for the classification layer part, as many other recent works do. In this setting, all tasks share the same classification layer that outputs probabilities for all classes for all tasks up to now. Hence, it needs to not only distinguish the classes within the task but also differentiate these classes from classes of other tasks.
> > >
> > > This is different from the multi-head architecture that applies a separate classifier to each task, which only needs to distinguish the classes within each task. The multi-head architecture was not commonly used in CL because it needs to know the task label for a test sample.
> > >
> > > * Lorenzo Bonicelli, Matteo Boschini, Angelo Porrello, Concetto Spampinato, and Simone Calderara. On the effectiveness of lipschitz-driven rehearsal in continual learning. Advances in Neural Information Processing Systems, 2022.
> > > * Matteo Boschini, Lorenzo Bonicelli, Pietro Buzzega, Angelo Porrello, and Simone Calderara. Class-
> > > incremental continual learning into the extended der-verse. IEEE Transactions on Pattern Analysis and Machine Intelligence, 2022.
> > > * R. Tiwari, K. Killamsetty, R. Iyer, and P. Shenoy. Gcr: Gradient coreset based replay buffer selection
> > > for continual learning. In 2022 IEEE/CVF Conference on Computer Vision and Pattern Recognition (CVPR), 2022.
> > > * Matteo Boschini, Lorenzo Bonicelli, Angelo Porrello, Giovanni Bellitto, Matteo Pennisi, Simone
> > > Palazzo, Concetto Spampinato, and Simone Calderara. Transfer without forgetting. In European
> > > Conference on Computer Vision, 2022.
> > > * Elahe Arani, Fahad Sarfraz, and Bahram Zonooz. Learning fast, learning slow: A general continual learning method based on complementary learning system. In International Conference on Learning Representations, 2022.
> > > * Mozhgan PourKeshavarzi, Guoying Zhao, and Mohammad Sabokrou. Looking back on learned experiences for class/task incremental learning. In International Conference on Learning Representations, 2022.
> > >
> > > **6. It would be helpful if the Authors can provide any measurement/hint has concerns what happens to the task accuracy during the stream of experiences.**
> > >
> > > Thanks for the suggestion! In the following tables, we report the average test accuracy of previous tasks at the end of each task during CL on two benchmarks, i.e., Seq-PathMNIST (class-IL) and Seq-PACS (domain-IL). The results show that $k$-FPF can always achieve the best performance among various CL methods at the end of any task during CL. Therefore, whenever one wants to apply the model during CL, $k$-FPF can achieve a model performing well on previous tasks.
> > >
> > > **The average accuracy of previous tasks at the end of each task during the training of CL on Seq-PathMNIST :**
> > > | Methods | Task 1 | Task 2 | Task 3 | Task 4 |
> > > |:------|:-----:|:---------:|:------:|:------:|
> > > |   $k$-FPF-CE | 99.95$\pm$0.04 | 95.41$\pm$1.98 | 81.92$\pm$2.26 | 76.72$\pm$1.94 |
> > > |   ER                | 98.62$\pm$1.59 | 83.06$\pm$3.12 | 74.60$\pm$3.18 | 57.54$\pm$3.05  |
> > > |   AGEM          | 99.71$\pm$0.19 | 46.58$\pm$3.13 | 36.12$\pm$3.17 | 27.33$\pm$3.93|
> > > |   iCaRL          | 99.98$\pm$0.02 | 86.86$\pm$5.47 | 66.62$\pm$5.64 | 57.67$\pm$1.13|
> > > |   FDR              | 99.97$\pm$0.06 | 48.06$\pm$0.82 | 55.75$\pm$6.55 | 40.08$\pm$4.13|
> > > |   DER              | 99.98$\pm$0.02 | 91.92$\pm$3.42 | 76.50$\pm$5.77 | 66.71$\pm$3.40 |
> > > |   DER++          | 99.95$\pm$0.06 | 94.06$\pm$6.14 | 80.35$\pm$3.32 | 71.09$\pm$2.60 |
> > >
> > > **The average accuracy of previous tasks at the end of each task during the training of CL on Seq-PACS :**
> > > | Methods | Task 1 | Task 2 | Task 3 | Task 4 |
> > > |:------|:-----:|:---------:|:------:|:------:|
> > > |   $k$-FPF-CE | 70.94$\pm$2.02 | 73.75$\pm$2.68 | 62.37$\pm$0.49 | 65.90$\pm$0.72 |
> > > |   ER                | 56.64$\pm$9.04 | 54.34$\pm$9.44 | 46.79$\pm$8.48 | 52.72$\pm$4.01  |
> > > |   AGEM          | 47.34$\pm$7.35 | 38.02$\pm$5.82 | 32.70$\pm$7.13 | 35.29$\pm$4.94  |
> > > |   FDR              | 58.59$\pm$4.36 | 54.00$\pm$4.01 | 46.38$\pm$4.80 | 48.50$\pm$4.67 |
> > > |   DER              | 48.49$\pm$9.40 | 45.28$\pm$8.88 | 34.48$\pm$7.81 | 47.63$\pm$3.85 |
> > > |   DER++          | 55.33$\pm$7.45 | 64.43$\pm$6.50 | 50.19$\pm$7.30 |  57.77$\pm$2.54 |

---

> > > > ### Comment · Reviewer_8gaT · 2022-11-20
> > > > **Thanks for the additional response**
> > > >
> > > > I appreciated the new comment, but let me make some point clearer as it seems there is not complete understanding of my points:
> > > > - About 1:  I am not stating this work is the same as [1] and [2]. I am saying this work discovers through an empirical method some aspects that were already known by the community under different assumptions, namely influence of last layer and batch norm parameters on forgetting.
> > > > - About 5: I am quite familiar on the setup of those papers and I am not talking multi-head here. I am saying that stating that no class-specific parameters exist is not correct as in CI, without multi-head output, you will anyway need to incrementally add class-specific outputs.
> > > >
> > > > This said, about 6, thanks very much: the results look quite interesting and strenghten the paper. My suggestion is to add them to the paper for all the datasets.
> > > >
> > > > About point 2: I appreciate popularity as an argument but when a claim of generality of a method is made, I prefer diversity. It is quite known that thing break down when considering CL with RNNs. In fact these are far less popular than CNNs in the community. Still, demonstrating to be able to isolate sensite parameter and control forgetting also in recurrent architectures would have added very much to the claim of generality.
> > > >
> > > > About point 3: I appreciate the little sensitivity but if I understand correctly your response you are confirming that the hyperparameters are chosen post-hoc, at the end, rather than in early stage. Correct?
> > > >
> > > > About point 4: I get the motivation about reducing the computational impact. Yet, it would be fairer to put the DER++ (and the other algorithms) in their SOTA conditions if a claim of superior performance is made. Otherwise, I suggest discussing a claim of good tradeoff between accuracy and computational complexity to make explicit that SOTA models are run under constraints rather than in their full form. Note that this is very important for reproducibility as in the future those reading and citing the results in this paper will need to know how to reference those results in their empirical analyses.
> > > >
> > > > Concluding, thanks very much for your discussion and prompt responses. I will keep these additional information in mind in the discussion and update my score accordingly.

---

> > > > > ### Author Response · Authors · 2022-11-20
> > > > > **Third Response to Reviewer 8gaT**
> > > > >
> > > > > Thank you for your prompt response! We appreciate your confirmation that some of your concerns are addressed, and our new experiments are "quite interesting and strengthen the paper." We also appreciate your clarification of some previous comments, which however could not fully support your justification of "lack of technical detail, reduced novelty and scarce generality of the results." In particular:
> > > > >
> > > > > **About point 1**:
> > > > > * This is about the "reduced novelty". However, the "some aspects" are not among the main contributions we claimed. **We do not make assumptions as [1] and [2]** that the influence of the last layer and batch norm parameters on forgetting are the largest. Instead, **we identify sensitive parameters solely based on the training dynamics**. Our observations are also different: the influence of the last layer and BN parameters are varying for different scenarios.
> > > > >
> > > > > * The studies in [1] and [2] are conducted in a different and less commonly used setting (pretrained backbone network frozen in CL). It is not clear how and when their results can be generalized to more common cases adopted by other CL works. In contrast, our studies are much more general since we adopt the most commonly used setting in CL and cover different architectures, different CL methods, different hyperparameters, different CL tasks (class-IL and domain-IL), etc.
> > > > >
> > > > > **About point 5**:
> > > > > * Thanks for the clarification! If so, **this is mainly a wording issue** and we are willing to provide a more precise description. It is obvious that the output classes are class-specific because different classes belong to different tasks. But each parameter or weight in the classification layer is trained for all classes rather than classes from a single task only (cross-entropy loss computed on all classes), as almost all previous CL methods do for class-IL, and we follow the same setting.
> > > > >
> > > > > * Note this is for class-IL only because it is the most popular setting studied in recent CL literature. Our paper also covers domain-IL, in which the classes for all tasks are exactly the same so they share exactly the same model, including the output nodes. There is a task-IL setting which assumes knowing the task boundary (in training) and task label (in test). It is much simpler and has already been solved. In Task-IL, the parameters in the classification layer are entirely task-specific.
> > > > >
> > > > > * Our methods (FPF and $k$-FPF) apply to both class-IL and domain-IL and outperform all baselines. This is definitely the opposite of "scarce generality of the results".
> > > > >
> > > > > **About point 6**: Thanks for your positive feedback on our new experiments! We will take your advice and add them to the paper for other datasets.
> > > > >
> > > > > **About point 2**: We agree that CL for RNN is a new topic rarely studied before and an open challenge worth studying in the future. In this paper, we cover different architectures, replay methods, hyperparameters, and CL settings that have been **widely studied in existing CL works** and our methods consistently bring improvements to all of them. This is generality, though not the ultimate generality as AGI that tries to cover all possible architectures. If not, all previous CL works not studying RNN should be rejected due to lack of generality, correct?
> > > > >
> > > > > **About point 3:**
> > > > > * **No, this is WRONG! The hyperparameter $p$ is determined before CL for most experiments**. For any model with BN, $p$ is preset to $97$. Otherwise, $p$ is preset to $70$. We only tuned $p$ in two experiments (with BN and w/o BN) and applied the result to all the other experiments.
> > > > > * In each experiment tuning $p$, we do not need to wait until the end but use the validation accuracy averaged over a few times of FPF in the early stage. Since the sensitivity and dynamic patterns of parameters are very stable during CL, rough grid search based on the early stage FPF accuracy suffices to find a good value for $p$.
> > > > >
> > > > > **About point 4: We already provided an analysis of the trade-off between accuracy and FLOPS in Fig. 4** for our methods and SOTA CL methods.
> > > > > FPF improves all CL methods by a large margin without notably extra computation. $k$-FPF consumes much less computation but achieves comparable performance as FPF.
> > > > >
> > > > > Concluding, though we can keep improving the paper to better address some of your concerns, e.g., adding experiments to the main paper and improving the wording about parameter sharing, we have addressed most of your concerns about novelty and generality, and there are few questions left. Based on the discussion, would you mind raising your rating to reflect the discussion results? Thanks!

---

> > > > > > ### Comment · Reviewer_8gaT · 2022-11-24
> > > > > > **New response**
> > > > > >
> > > > > > Thanks for the additional comments. I certainly hoped point 5 was a wording issue because I could not see any reasonable CI setting without classes-specific outputs.
> > > > > >
> > > > > > This said, I respectfully disagree that trying different hyperparameters and CI/DI settings results in an exceptional level generality in the CL community. SOTA works need to show evidence of working adequately at least in CI/DI settings: don't see anything out of the ordinary in this contribution. Again, since the focus here is on claiming that general methods are provided to investigate parameters sensitivity and consequentely control forgetting, it seems to me a stronger result to be able to show that the method generalizes over diverse neural paradigms, rather than over variations of the same architecture.
> > > > > >
> > > > > > About 3: I appreciate the clarification. Now: how do I choose p for a new architecture? If p is not chosen in model selection, how is this set? Because I keep re-reading appendix A-1 and it is not clear to me what is the procedure to set p without model selection, nor what is the procedure used to determine p=97 for one architecture and p=70 for the other one.
> > > > > >
> > > > > > About 4: my point is that Fig. 4 shows the accuracy-effort tradeoff where the accuracy of related works is not optimized. In that picture (as in the rest of the results) DER++ (as other algorithms) should be put in conditions to reach its SOTA results (and possibly show an even heavier computational cost, is so happens to be).

---

> > > > > > > ### Author Response · Authors · 2022-11-27
> > > > > > > **Fourth Response to Reviewer 8gaT**
> > > > > > >
> > > > > > > **About 5: I certainly hoped point 5 was a wording issue because I could not see any reasonable CI setting without classes-specific outputs.**
> > > > > > >
> > > > > > > * Please read our description of the classification layer used in our paper as well as most existing CL papers (we repeated it several times): it is a simple fully connected layer and every output node corresponds to a class of a task from the sequence.
> > > > > > >
> > > > > > > **I respectfully disagree that trying different hyperparameters and CI/DI settings results in an exceptional level generality in the CL community.**
> > > > > > >
> > > > > > > * Trying different parameters, CI/DI settings, and most widely used architectures is not an exceptional level generality, but it also cannot result in a conclusion of "scarce generality of the results".
> > > > > > >
> > > > > > > * **What is really exceptional about our method are** (we repeated them several times): (1) FPF can be generally applied to any existing CL methods and consistently improve their performance; (2) $k$-FPF entirely removes the every-step replay strategy used in a majority of recent CL methods. This saves a great amount of computation, avoids the bias of every-step replay, and achieves better performance.
> > > > > > >
> > > > > > > **It seems to me a stronger result to be able to show that the method generalizes over diverse neural paradigms, rather than over variations of the same architecture.**
> > > > > > >
> > > > > > > * This is not true. We validated our methods on MLP and CNNs, which are very different architectures. Moreover, we validated our method on VGG and ResNet, which are two variations of CNNs and most researchers would agree they are two very different variations. **They cover all architectures used in most mainstream CL papers.** Including more architectures will obviously bring new contributions and additional novelty but our focused problem here is not to extend current CL methods to more architectures (it is another new paper).
> > > > > > >
> > > > > > > **About 3: I appreciate the clarification. how do I choose p for a new architecture? what is the procedure used to determine p=97 for one architecture and p=70 for the other one.**
> > > > > > >
> > > > > > > * Thanks for acknowledging our clarification! We would like to repeat our previous response: for a new architecture, $p$ is chosen in the early stage of CL using the validation accuracy averaged over a few times of FPF (in $k$-FPF), following the method described in Point 1 of "Response to all reviewers".
> > > > > > >
> > > > > > > * We chose $p=97$ and $p=70$ for two broad classes of models, i.e., the ones with BN and the ones without BN, respectively. They are NOT two specific architectures. We achieved each value by tuning $p$ (rough grid search using the above method) on only one architecture from that class and applied it to all architectures of the same class. Our experiments show that the chosen $p$ performs well over different architectures of the same class on different datasets and CL settings when combined with different CL methods.
> > > > > > >
> > > > > > > **About 4: my point is that Fig. 4 shows the accuracy-effort tradeoff where the accuracy of related works is not optimized. DER++ should be put in conditions to reach its SOTA results .**
> > > > > > >
> > > > > > > * Thanks and we report the requested comparison in the table below, which compares the accuracy and FLOPs of our methods with the original results in DER [1] when allowing a large number of epochs on the same data for each task. In both class-IL and domain-IL, k-FPF-CE is comparable to DER++ and k-FPF-KD is better than DER++ on the accuracy but spends much less FLOPs. These results demonstrate that our methods can outperform SOTA methods in various scenarios.
> > > > > > >
> > > > > > >
> > > > > > > | Methods  | Seq-CIFAR-10 Accuracy  | Seq-CIFAR-10 FLOPs (B)  |  R-MNIST Accuracy | R-MNIST FLOPs (B)
> > > > > > > |:---|:---:|:---:|:---:|:---:|
> > > > > > > | **$k$-FPF-CE**          | 71.93$\pm$0.58 | 9208.85 | 91.15$\pm$0.29 | 0.64  |
> > > > > > > | **$k$-FPF-KD**          | **74.32$\pm$0.32** | 9208.85 | **93.61$\pm$0.45** | 0.64|
> > > > > > > |  DER                           | 70.51$\pm$1.67 | 16726.26 | 92.24$\pm$1.12 | 1.29|
> > > > > > > |  DER++                      | 72.70$\pm$1.36 | 25089.39 | 92.77$\pm$1.05 | 1.93|
> > > > > > >
> > > > > > > [1] Pietro Buzzega, Matteo Boschini, Angelo Porrello, Davide Abati, and Simone Calderara. Dark experience for general continual learning: a strong, simple baseline.

---

> > > ### Author Response · Authors · 2022-11-20
> > > **New Response to Reviewer 8gaT (1)**
> > >
> > > **1. The claimed novelty in being the first paper in studying the sensitivity of parameters in CL is not entirely supported to the extent of my knowledge. Most of the empirical findings of this experiment are known from previous works.**
> > >
> > > * **This is either a misinterpretation of our paper or the works you mentioned**. They do not compare the sensitivity or importance of different parameters in CL. Their studied problems, proposed methods, empirical findings, and their major contributions are entirely different from ours:
> > >
> > > * [1] studies the functionality of different types of **output layers** when freezing the pretrained backbone network. Our setting and problem are obviously different because our study does not start CL from a pretrained model and we study the sensitivity of parameters from all layers from the first to the last. Our findings are also different: For domain-IL or models with BacthNorm, the last layer is no longer the most sensitive group of parameters, but other parameters like BatchNorm and some convolution layers are more sensitive to distribution shift.
> > >
> > > * Similar to [1], [2] also uses a pre-trained backbone network and freezes a large number of convolutional parameters during CL. They propose to replace Batch Normalization with Batch Renormalization, which depends on the global mean and variance to mitigate forgetting. In contrast, we do **not use any presumption** of the sensitive parameters. We determine the parameters to finetune based on a data-driven analysis. For different neural nets, settings (class-IL or domain-IL), and datasets, the sensitive parameters can vary and our method is more adaptive in identifying them.
> > >
> > > **2. The paper only considers an MLP and two instances of a CNN; I would expect to see a larger choice of models being assessed.**
> > >
> > > Our experiments cover different types of models used in most CL papers. In a variety of recent works (for example, the paper listed below) in the CL community, ConvNets with BN such as ResNet and MLP are the most popular neural networks studied in different CL scenarios and benchmarks.
> > > * Fei Ye and Adrian G. Bors. Task-free continual learning via online discrepancy distance learning.
> > >  Advances in Neural Information Processing Systems, 2022.
> > > * Alex Ororbia, Ankur Mali, C. Lee Giles, and Daniel Kifer. Lifelong neural predictive coding: Learning cumulatively online without forgetting.  Advances in Neural Information Processing Systems, 2022.
> > > * Qing Sun, Fan Lyu, Fanhua Shang, Wei Feng, and Liang Wan. Exploring example influence in continual learning.
> > > Advances in Neural Information Processing Systems, 2022.
> > > * Hyundong Jin and Eunwoo Kim. Helpful or harmful: Inter-task association in continual learning.
> > >  Computer Vision – ECCV 2022, pp. 519–535, Cham, 2022.
> > > * Arjun Ashok, K. J. Joseph, and Vineeth N. Balasubramanian. Class-incremental learning with cross-space clustering and controlled transfer. Computer Vision – ECCV 2022, pp. 105–122,
> > > Cham, 2022.
> > > * Qingsen Yan, Dong Gong, Yuhang Liu, Anton van den Hengel, and Javen Qinfeng Shi. Learning
> > > bayesian sparse networks with full experience replay for continual learning. In 2022 IEEE/CVF
> > > Conference on Computer Vision and Pattern Recognition (CVPR), pp. 109–118, 2022.
> > >
> > > **3. But if that early stage assessment needs an hyperparameter chosen by looking at the aggregated performance at the end of the training experiences, then it is not much early-stage.**
> > >
> > > **The early-stage assessment is not sensitive to hyperparameters**. As shown in Fig. 1-2, the sensitivity and dynamic patterns of different parameters (e.g., their ranking) do not change drastically under different scenarios such as task changing or varying buffer sizes. So the aggregated performance at the end is not necessary to identify the sensitive parameters.
> > >
> > > **4. Instead I see that DER++ results are lower: do you have any hint of why this happens as the network backbone seems the same.**
> > >
> > > * Yes, we use the same network backbone as Buzzega et al, which is ResNet-18.
> > > * The accuracy of DER++ in our paper is lower than the original paper because we use **much fewer** training epochs ($5$ epochs) for each task than the DER paper, which repeats training on the same training samples of each task for $50$ or $100$ epochs. We did not adopt $50$ or $100$ epochs since they are much larger than what has been used in previous works. They are also impractical for streaming data and edge devices with limited computational speed.
> > > * Nevertheless, on Seq-Tiny-ImageNet, our method with only $5$ epoch per task achieves higher accuracy ($20.56±0.32$) than the accuracy ($19.38±1.41$) achieved in Buzzega et al. even using much more epochs.
> > >
> > > [1] Lesort, T., George, T. \& Rish, I. Continual Learning in Deep Networks: an Analysis of the Last Layer. arXiv (2021)
> > >
> > > [2] Lomonaco, V., Maltoni, D. \& Pellegrini, L. Rehearsal-Free Continual Learning over Small Non-I.I.D. Batches. in CVPR Workshop on Continual Learning for Computer Vision 246–247 (2020)

---

> ### Author Response · Authors · 2022-11-30
> **Thank you for your review!**
>
> Many thanks again for your precious time inspecting our work and valuable comments that helped improve our work a lot.
> We have meticulously clarified all of the concerns that you have raised.
>
> We would appreciate it if you could take a moment to re-evaluate our revised paper in light of our response and find our improved work more positive!
>
> If there are any clarifications you need further, we are always open and delighted to discuss with you.

---

> ### Author Response · Authors · 2022-12-06
> **Looking forward to a further discussion before the deadline**
>
> Thanks again for your great efforts in reviewing our paper!
>
> We have addressed all your questions in detail. As the deadline for the discussion is fast approaching, we are really looking forward to having a further discussion. Would you mind checking our response and letting us know if you have further questions?
>
> We sincerely encourage the reviewer to revise the recommendation score based on our updated version, new results and clarifications.
>
> Thank you!

---

### Official Review · Reviewer_hdv9 · 2022-10-24

**Confidence:** 4
**Correctness:** 4
**Technical Novelty And Significance:** 3
**Empirical Novelty And Significance:** 3
**Recommendation:** 8

**Clarity, Quality, Novelty And Reproducibility:**

Overall, the paper is well-written and clear. In terms of novelty and quality, I believe the contributions are very limited. Finally, the authors provide details of their experimental setup and hyper-parameters used to reproduce the results.

**Strength And Weaknesses:**

### Strengths


- The paper is well-written and easy to follow.


- The direction of understanding network dynamics in continual learning is very important.


- The proposed method is simple and intuitive and performs well.



### Weaknesses


- First, I should say that I am not sure if the drift in parameters is the best perspective for measuring the dynamics of the layers. For instance, in [2], the authors show that small/large euclidean parameter distance does not necessarily correspond to high/low forgetting, which can depend on many other factors. In other words, you given parameters $\theta$ at the end of task 1, you can simply find $\theta_1$ and $\theta_2$ where $| \theta - \theta_1 |  < | \theta - \theta_2 | $ but the performance of $\theta_2$ is better than $\theta_1$.



- In addition, as the authors mention in Appendix A.4, some of the presented results (e.g., final blocks being more sensitive) are well-established in the literature [1]. While this work dives a little bit deeper into designing FPF and measuring the compute/performance trade-offs, still, the novelty and contribution of this work are limited.





**References**:
[1] Ramasesh, Vinay Venkatesh et al. “Anatomy of Catastrophic Forgetting: Hidden Representations and Task Semantics.” ICLR 2021.
[2] Mirzadeh, Seyed Iman, et al. "Linear Mode Connectivity in Multitask and Continual Learning. ICLR 2021.




**Summary Of The Paper:**


The paper studies the dynamics of different layers in continual learning scenarios, mainly by measuring the shift in parameters. The authors show that only a small subset of parameters (e.g., final FC layer, BN stats) are very sensitive to data drift. To exploit this observation, the authors propose forgetting prioritized finetuning (FPF), which only finetunes a small subset of parameters. In addition, they propose k-time FPF, which replaces the every-step experience replay.

**Summary Of The Review:**

Overall, I believe this paper is a borderline paper. Although I believe the contributions of this work is limited, I pretty much enjoyed the direction of understanding NN dynamics in CL, which is done rarely in the literature.

**Post-Rebuttal Update:**
I want to thank the authors for their response. Given that the authors have addressed most of my concerns, I would like to increase the score.

---

> ### Author Response · Authors · 2022-11-17
> **Response to Reviewer hdv9**
>
> Thanks for your comments and your efforts for reviewing our paper! In the following, we provide answers to your questions.
>
> **1. I should say that I am not sure if the drift in parameters is the best perspective for measuring the dynamics of the layers. In [2], the authors show that small/large euclidean parameter distance does not necessarily correspond to high/low forgetting.**
>
> A significant difference between our study and [2] is that we only measure the parameter difference between two close steps on the optimization trajectory and focus on studying its dynamics. Since the models of the two steps are close to each other, Euclidean or straight line distance between them is relatively accurate. In contrast, [2] used the Euclidean distance to measure the difference between models of two distant steps (e.g., task-1 and task-5), which can be inaccurate. Their study is mainly based on the distance rather than its training dynamics. Hence, the conclusions can be different.
>
> **2. In addition, as the authors mention in Appendix, some of the presented results (e.g., final blocks being more sensitive) are well-established in the literature [1].**
>
> Our analysis and conclusion for our empirical study are **entirely different** from those in [1]:
> [1] observed that freezing bottom layers had little impact on the performance of the second task, so it concludes that the final/top blocks are more sensitive. But [1] did not find that different modules' parameters show different sensitivities to task/domain shift in CL.
>
> Instead, we compare the sensitivity of parameters belonging to different types of modules and layers. For example, we found that convolution and batch-norm parameters in the bottom layers are much more sensitive than any others. Since the variance of parameters' sensitivity can be very high within a layer/block, it is less informative to simply compare the sensitivity across layers.
>
> **3. While this work dives a little bit deeper into designing FPF and measuring the compute/performance trade-off, still, the novelty and contribution of this work are limited.**
>
> Our design of FPF and $k$-FPF is based on novel empirical studies different from previous works in two ways: (1) studying training dynamics of parameters in CL; (2) comparison between different types of modules and layers.
>
> The main novelty of FPF is that it only selects a tiny portion of sensitive parameters for finetuning, and the main contribution is that adding such a simple and efficient finetuning at the end can significantly improve any existing CL method on the accuracy.
>
> The main novelty and impact of $k$-FPF are that it totally removes the every-step replay, which doubles the training cost (at least) and is a ``must-have'' of most CL methods proposed in the past three years.
>
> [1] Ramasesh, Vinay Venkatesh et al. “Anatomy of Catastrophic Forgetting: Hidden Representations and Task Semantics.” ICLR 2021.
>
> [2] Mirzadeh, Seyed Iman, et al. "Linear Mode Connectivity in Multitask and Continual Learning. ICLR 2021.

---

> ### Author Response · Authors · 2022-12-08
> **Thank you for your review!**
>
> We are glad to see that your concerns have been addressed after the rebuttal. Thank you for the careful re-evaluation and for recognizing the contribution of our work.

---

### Official Review · Reviewer_LsHP · 2022-10-25

**Confidence:** 5
**Correctness:** 3
**Technical Novelty And Significance:** 3
**Empirical Novelty And Significance:** 4
**Recommendation:** 6

**Clarity, Quality, Novelty And Reproducibility:**

The paper is somewhat clear, but some notations are confusing. Though the methods are simple and might not be novel, showing the sensitive region of neural network in CL can be novel.

**Strength And Weaknesses:**

**Pros:**

P1. FPF is simple and highly effective, and showing sensitive regions in the parameters of neural network in CL can be a novel finding.

P2. FPF can be easily plugged-in baselines, and it increases the performance significantly.

**Cons:**

C1. In Section 4.1, the notation in the denominator of the metric for the change of parameters is somewhat confusing. Is it $\theta_{\ell,n}^{t+1}$? If it is, is it feasible to use this metric for measuring the dynamics of parameters?

C2. Selecting the sensitive part of the neural network is little bit confusing. How can we select the parts when we use much larger architectures (e.g. ResNet-110)? It would be better to specify the standard for selecting the sensitive region.

C3. There is no comparison between FFP and the method that fine-tune all parameters using memory buffer. I think the results are quite different from "ER". It would be better to show the comparison between two methods with respect to the FLOPs and the average accuracy.

C4. There is no experiments on the task sequence containing totally different datasets (e.g. CIFAR-100 & MNIST). The trends that Conv & FC layers are sensitive would be different from the experiments in the paper.

**Summary Of The Paper:**

This paper proposes a simple technique for preventing the catastrophic forgetting problem in continual learning (CL). By showing that there are sensitive parts in neural network model when learning a novel task, the proposed approach (FPF) just fine-tune the sensitive part of a model after learning sequence of tasks. Furthermore, the additional computation cost for performing FPF the sensitive part is somewhat marginal. In the experiments, authors show that applying FPF can drastically increase the performance of baselines.

**Summary Of The Review:**

I vote to marginally accept this paper. Though some experiments are missing, and the proposed solutions are somewhat straightforward, finding out which parts are sensitive to the distribution shift in CL has significant contribution.

---

> ### Author Response · Authors · 2022-11-17
> **Response to Reviewer LsHP**
>
> Thanks for your comments! Please find our new experiments suggested by you and our answers to your questions below.
>
> **1. In Section 4.1, the notation in the denominator of the metric for the change of parameters is somewhat confusing. Is it $\theta^{t+1}_{l,n}$? If it is, is it feasible to use this metric for measuring the dynamics of parameters?**
>
> The denominator is $|\theta_\ell|$, which denotes the number of parameters in $\theta_\ell$. This metric computes the average difference of parameters in $layer-\ell$.
>
> It aims to measure the change of parameters over a few steps on the optimization trajectory, along which Euclidean distance between consecutive steps provides an accurate measurement of their difference. But Euclidean distance can be problematic when applied to two arbitrary models due to the redundancy and symmetry of parameters.
>
> **2. Selecting the sensitive part of the neural network is a little bit confusing. It would be better to specify the standard for selecting the sensitive region.**
>
> Please refer to our reply to Question 1 in Response to all reviewers.
>
> **3. There is no comparison between FPF and the method that fine-tune all parameters using memory buffer.**
>
> In the following table, we compare FPF with FPF-ALL (which finetunes all parameters) when applied to different CL methods for two types of CL, i.e., class-IL and domain-IL. The results show that FPF consistently achieves comparable or slightly higher accuracy than FPF-ALL by spending significantly fewer FLOPs. This demonstrates the advantage of FPF on efficiency.
>
> | Methods  | Seq-PathMNIST Accuracy  | Seq-PathMNIST FLOPs (B)  |  Seq-PACS Accuracy | Seq-PACS FLOPs (B)
> |:---|:---:|:---:|:---:|:---:|
> | **$k$-FPF-SGD**           | 76.72$\pm$1.94  | 21.35  | 65.90$\pm$0.72  |  148.25  |
> | **$k$-FPF-ALL-SGD**   |  75.74$\pm$2.91 |43.95 | 64.48$\pm$2.23 | 174.60|
> |  **FPF**+ER                    |  69.83$\pm$2.87 | 4.68 | 64.27$\pm$1.91 | 24.39
> |  **FPF-ALL**+ER            | 70.64$\pm$4.00 | 8.79 | 63.81$\pm$2.33 | 34.92
> |  **FPF**+AGEM              |  73.32$\pm$3.73 | 7.07 | 62.40$\pm$1.89 | 18.47
> |  **FPF-ALL**+AGEM     |  74.80$\pm$3.12 | 8.79 | 62.65$\pm$1.65 | 34.92
> |  **FPF**+iCaRL              |  74.35$\pm$4.89 | 4.27 | -              | -
> |  **FPF-ALL**+iCaRL     |  72.77$\pm$4.12 | 8.79 | -              | -
> |  **FPF**+FDR               |  75.59$\pm$2.64 | 2.94 | 65.47$\pm$1.13 | 11.70
> |  **FPF-ALL**+FDR        | 74.24$\pm$1.48 | 8.79 | 64.88$\pm$2.28 | 34.92
> |  **FPF**+DER               |  74.80$\pm$3.45 | 2.96 | 65.69$\pm$1.66 | 18.47
> |  **FPF-ALL**+DER       |  74.54$\pm$3.19 |8.79 | 66.22$\pm$0.87 | 34.92
> |  **FPF**+DER++           |  77.37$\pm$1.32 | 4.68 | 66.89$\pm$1.32 | 24.39
> |  **FPF-ALL**+DER++    |  77.16$\pm$1.45 | 8.79 | 65.19$\pm$1.33 | 34.92
>
> **4. There are no experiments on the task sequence containing totally different datasets.**
>
> Since we compared our method with existing CL methods, we used exactly the standard benchmarks adopted by most published CL methods, each of which extracts different tasks from the same dataset.
>
> In order to answer your question, we concatenate the CL tasks from two datasets (i.e., Seq-CIFAR-10 and Seq-PathMNIST) to form a twice-longer task sequence and evaluate the training dynamics of different groups of parameters.
>
> The result is shown in  Fig.7 in the Appendix of the revised paper. The vertical dashed line at epochs $=30$ is the boundary between the two datasets.
> Although the two datasets are from different domains and thus have different distributions, the sensitivity of parameters stays consistent with our observations on tasks from a single dataset. This indicates that the sensitive parameters almost do not change across different tasks/datasets, so they can be identified at very early stages.

---

> > ### Comment · Reviewer_LsHP · 2022-12-08
> > **Response to the comments**
> >
> > Sorry for the late reply.
> >
> > After reading all the comments, I think the problems are well addressed.
> >
> > Though reviewer 8gaT voted to reject this paper due to the two major reasons mentioned in the official comment, I still vote to accept this paper because of the novel empirical findings on the presence of the sensitive part in CNN or MLP in continual learning setting and a simple solution leveraging this findings.
> >
> > Therefore, I keep my original score.

---

> > > ### Author Response · Authors · 2022-12-08
> > > **Thank you for your reply**
> > >
> > > Thanks for your reply and positive feedback! We greatly appreciate your careful consideration of our responses and revision, as well as our discussion with other reviewers.
> > >
> > > We are glad to hear that all your concerns have been addressed by our response. We believe our study, method, and insights can contribute a new idea generally helpful to a variety of researchers in the community. To make this happen, your support is critical! Hence, we humbly hope you can increase the score.
> > >
> > > We believe that our detailed response and new experiments address the major concerns of most reviewers and re-validate our novelty. Based on these, Reviewer hdv9 has raised the score from 6 to 8. Would you mind considering to raise your score as well?
> > >
> > > We are looking forward to hearing you. Thanks a lot!

---

> ### Author Response · Authors · 2022-11-30
> **Thank you for your review!**
>
> We sincerely appreciate your time and effort in reviewing our paper! We believe your constructive comments will further strengthen our paper, especially the comparison between FPF and the method that fine-tune all parameters.
>
> In the response and revised paper, we have made every effort to address your concerns. Particularly,
> * A clearer description of selecting the sensitive part of different neural networks.
> * A new experiment of comparing FPF with the method that fine-tune all parameters.
> * A new experiment of studying the training dynamics of parameters on the task sequence containing totally different datasets.
>
> Hence, it would be highly appreciated if you could provide feedback to our responses or confirm whether there is no remained concern. If you have any further concerns, questions, or suggestions, we are willing to discuss and reflect on them in the next revision. Thank you!

---

> ### Author Response · Authors · 2022-12-06
> **Looking forward to a discussion before the deadline**
>
> Thanks again for your great efforts in reviewing our paper!
>
> We have addressed all your questions in detail. As the deadline for the discussion is fast approaching, we are really looking forward to having a discussion with you on the OpenReview system. Would you mind checking our response and letting us know if you have further questions?
>
> We would appreciate it if you could take a moment to re-evaluate our revised paper in light of our response and find our improved work more positive!
>
> Thank you!

---

### Author Response · Authors · 2022-11-17
**Response to all reviewers**

**1. How to select the sensitive part of the neural networks?**

We select sensitive parameters according to their training dynamics in the early epochs. Examples of the training dynamics for different layers are shown in Fig 1-2, and their ranking does not change over epochs.
Specifically, we sort layers by their training dynamics values in descent order. Then we greedily add layers one after another to the sensitive group until the sum of all selected layers' training dynamics exceeds $p$ percent of the sum for all layers, where $p$ is a hyper-parameter.

In our experiments, for models with the batch-norm layers like ResNet-18, FPF and $k$-FPF outperform all baselines when $p = 97$, when only $18.90$% of parameters in the neural network are regarded as sensitive parameters.
For other models like MLP and VGG-11, $p = 70$ and only $1.12$% and $0.32$% of parameters are regarded as sensitive parameters.

**2. The novelty of this paper.**

The studied problem is novel: we are the first work to study the sensitivity of parameters in CL for diverse types of neural nets. The proposed CL strategy is novel: $k$-FPF firstly makes the every-step replay unnecessary, which is a technique dominating the past three years' CL research but requires much more computation than $k$-FPF.

Our proposed technique is simple and intuitive, i.e., selecting sensitively varying parameters for fine-tuning effectively reduces forgetting. However, this technique has NOT been previously explored in CL.

Our FPF method is complementary to most existing CL methods and consistently improves their performance. Moreover, we achieve the new best performance among all the baselines, as shown in the experiments.

Our $k$-FPF method fundamentally removes the costly every-step replay in most of the current CL methods and replaces them with occasional FPF. This is a significant novelty leading to better efficiency of CL.

---

### Decision · Program_Chairs · 2023-01-20

**Decision:**

Reject

**Justification For Why Not Higher Score:**

- The empirical study does not provide very novel findings as they were either reported in previous works or are already known to most researchers and practitioners.
- The proposed method of selecting the hyperparameter p to decide which parameters to fine-tune and not does not seem principled, and the paper lacks any algorithms or detailed descriptions to properly implement it and reproduce the results.

**Justification For Why Not Lower Score:**

N/A

**Metareview: Summary, Strengths And Weaknesses:**

This paper studies which parameters in deep neural networks are susceptible to catastrophic forgetting in a continual learning scenario, and provide findings that only a small portion of the parameters are sensitive to forgetting. Specifically, the most sensitive layers for MLP were the top layers, and for VGG, they were the convolutional layers, and for ResNet, they were the batch normalization layers. Based on this finding, the authors propose forgetting prioritized finetuning, which fintunes only those most sensitive layers, which can be also applied k-times during the course of training with any continual learning algorithms. The experimental results with diverse baselines on multiple class-incremental and domain-incremental benchmarks show the effectiveness of the proposed approach.

The paper initially received split reviews, and fell into a borderline case. The reviewers considered the tackled problem as important, and were positive about some of the novel observations (sensitivity of the conv layers in CNNs), general applicability of the proposed forgetting prioritized finetuning, extensive experimental validation and the impressive performance obtained using the proposed method.

However, the reviewers were also concerned with the following weaknesses:
- Limited novelty in the empirical findings as they are either trivially known (the upper layers are more task-specific), or are already reported in previous works [1][2][3].
- Questionable reproducibility of the proposed forgetting prioritized finetuning due to the lack of details (e.g. pseudo-code of the algorithm, how the hyperparameters have been selected).

This led to in-depth discussions between the authors and reviewers, as well as among reviewers.

In the final zoom meeting, the reviewers reached an agreement to recommend a rejection, despite some of the novel findings from the paper, as they reached the following conclusions
- Most of the findings provided in the paper are already known to most researchers and practitioners (continual learning has unequal impact on the parameters, and it has larger impact on the upper, task-specific layers, or batch normalization layers that are sensitive to the task distribution).
- The analysis of impact of continual learning on each layer has been done in [3], for the case of pretrained language models.
- The finding that some of the earlier convolutional layers are impacted more by continual learning is somewhat novel, but the authors did not perform any in-depth analysis of why they are more sensitive to forgetting.
- The current descriptions of the proposed method in appendix A.1 and the experimental settings in the experimental section are insufficient to understand the method or reproduce the results.
- The proposed method does not provide a principled way to select the important hyperparameter p, which are set to arbitrary values for each architectures (e.g. p=97 for ResNet-19 and p=70 for MLP and VGG-11).
- The fairness of the experiments and the practicality of the proposed method as the selection of the hyperparameters are done using grid search and the authors reported the best performance of all combinations of hyperparameters.

Based on these discussions, the reviewers reached a consensus that the paper should undergo a major revision, especially focusing on improving the clarity regarding the method and the experiments section, as the current method does not appear principled and the experimental results do not seem reproducible from the current paper. As for the empirical analysis, it would be better if the authors further investigate the causes of the inequalities that are observed as the current findings do not provide much useful insights.

[1] Understanding the Role of Training Regime in Continual Learning, NeurIPS 2020
[2] Anatomy of Catastrophic Forgetting: Hidden Representations and Task Semantics, ICLR 2021
[3] Pretrained Language Model in Continual Learning: A Comparative Study, ICLR 2022



















**Summary Of Ac-Reviewer Meeting:**

However, the reviewers were also concerned with the following weaknesses:
- Limited novelty in the empirical findings as they are either trivially known (the upper layers are more task-specific), or are already reported in previous works [1][2][3].
- Questionable reproducibility of the proposed forgetting prioritized finetuning due to the lack of details (e.g. pseudo-code of the algorithm, how the hyperparameters have been selected).

This led to in-depth discussions between the authors and reviewers, as well as among reviewers.

In the final zoom meeting, the reviewers reached an agreement to recommend a rejection, despite some of the novel findings from the paper, as they reached the following conclusions
- Most of the findings provided in the paper are already known to most researchers and practitioners (continual learning has unequal impact on the parameters, and it has larger impact on the upper, task-specific layers, or batch normalization layers that are sensitive to the task distribution).
- The analysis of impact of continual learning on each layer has been done in [3], for the case of pretrained language models.
- The finding that some of the earlier convolutional layers are impacted more by continual learning is somewhat novel, but the authors did not perform any in-depth analysis of why they are more sensitive to forgetting.
- The current descriptions of the proposed method in appendix A.1 and the experimental settings in the experimental section are insufficient to understand the method or reproduce the results.
- The proposed method does not provide a principled way to select the important hyperparameter p, which are set to arbitrary values for each architectures (e.g. p=97 for ResNet-19 and p=70 for MLP and VGG-11).
- The fairness of the experiments and the practicality of the proposed method as the selection of the hyperparameters are done using grid search and the authors reported the best performance of all combinations of hyperparameters.